# Conformal Prediction Beyond the Seen: A Missing Mass Perspective for Uncertainty Quantification in Generative Models

**Sima Noorani**[*]
University of Pennsylvania
nooranis@seas.upenn.edu

**Shayan Kiyani**[*]
University of Pennsylvania
shayank@seas.upenn.edu

**George Pappas**
University of Pennsylvania
pappasg@seas.upenn.edu

**Hamed Hassani**
University of Pennsylvania
hassani@seas.upenn.edu

## Abstract

Uncertainty quantification (UQ) is essential for safe deployment of generative AI models such as large language models (LLMs), especially in high-stakes applications. Conformal prediction (CP) offers a principled uncertainty quantification framework, but classical methods focus on regression and classification, relying on geometric distances or softmax scores–tools that presuppose structured outputs. We depart from this paradigm by studying CP in a query-only setting, where prediction sets must be constructed solely from finite queries to a black-box generative model, introducing a new trade-off between coverage, test-time query budget, and informativeness. We introduce *Conformal Prediction with Query Oracle* (CPQ), a framework characterizing the optimal interplay between these objectives. Our finite-sample algorithm is built on two core principles: one governs the optimal query policy, and the other defines the optimal mapping from queried samples to prediction sets. Remarkably, both are rooted in the classical *missing mass problem* in statistics. Specifically, the optimal query policy depends on the rate of decay–or the derivative–of the missing mass, for which we develop a novel estimator. Meanwhile, the optimal mapping hinges on the missing mass itself, which we estimate using Good-Turing estimators. We then turn our focus to implementing our method for language models, particularly in open-ended LLM tasks involving question answering, multi-step reasoning, and structured information extraction, where outputs are vast, variable, and often under-specified. Fine-grained experiments[2] on three real-world open-ended tasks and two LLMs, show CPQ's applicability to *any black-box LLM* and highlight: (1) individual contribution of each principle to CPQ's performance, and (2) CPQ's ability to yield significantly more informative prediction sets than existing conformal methods for language uncertainty quantification.

## 1 Introduction

Generative models such as LLMs and diffusion models are widely deployed in high-stakes applications, yet they often produce unreliable outputs. These models may generate plausible but incorrect information, hallucinate facts, or exhibit inconsistency across runs [1–4]. Uncertainty quantification (UQ) is therefore essential for safe and trustworthy deployment of generative AI, enabling downstream users to detect unreliable outputs and make informed decisions under uncertainty.

---

[*]Equal contribution. Correspondence to: nooranis@seas.upenn.edu, shayank@seas.upenn.edu.

[2]We release our code at https://github.com/nooranisima/CPQ-missing-mass

39th Conference on Neural Information Processing Systems (NeurIPS 2025).

Conformal prediction (CP) is a statistical framework for UQ in supervised learning [5–7], where input-output pairs $(X, Y)$ are drawn from an unknown distribution. Instead of a single prediction, CP produces *prediction sets* calibrated to include the true label with high probability. Formally, given a miscoverage level $\alpha \in (0, 1)$, CP guarantees $\mathbb{P}(Y \in C(X)) \geq 1 - \alpha$, where $C(X)$ is the prediction set for input $X$. This holds under minimal assumptions: CP is *distribution-free* and *model-agnostic*, making it widely applicable. These properties have made CP a key tool in deploying ML systems in high-stakes settings. Recent work also shows that CP sets are essential for *risk-sensitive decision making*, where decisions must account for predictive uncertainty in a principled way [8].

CP has been extensively studied for classical tasks such as classification and regression [9–12]. In these settings, uncertainty is typically expressed through prediction sets of the form $\{y : S(x, y) \leq q\}$, where $S(x, y)$ is a nonconformity score measuring how atypical a label $y$ is for a given input $x$, and $q$ is a calibrated threshold. In regression, $S(x, y)$ might be $|y - f(x)|$, where $f(x)$ is a trained model. In classification, the score is often based on softmax probabilities, such as $1 - f_y(x)$, where $f_y(x)$ is the predicted probability for class $y$. However, this approach does not directly carry over to generative modeling—such as open-ended text generation—where outputs come from an immense, unstructured space of discrete sequences. While one can define similarity metrics over text or images, the core difficulty lies not in the absence of a distance, but in the fact that sets defined via these distances—such as "all outputs within a radius of $q$"—are typically intractable and hard to represent. In generative models like LLMs, the model does not expose a full probability distribution over the output space, but instead only provides a *query oracle*—a mechanism for sampling one output at a time. These challenges motivate the question: *Can we design conformal prediction procedures that meaningfully quantify uncertainty when the model only provides samples of its output space?*

Recent works have made progress toward adapting CP to query-based generative models [13, 14]. However, two key challenges remain mainly unresolved. First, querying at test time is resource intensive–more queries improve output exploration but incur substantial computational cost. Second, users often seek uncertainty quantification at high coverage levels (e.g., $90\%$), even when the model's few-shot accuracy is much lower (e.g., $60 - 70\%$). In such regimes, some prediction sets are necessarily non-informative—effectively suggesting that the true output could be anything—because the model fails to produce it within the query budget. These challenges highlight a fundamental trade-off between coverage, informativeness, and test-time query cost. Our goal is to design conformal procedures that navigate this trade-off by *minimizing the number of non-informative prediction sets while maintaining valid coverage under a limited query budget*.

A central insight in addressing this challenge is recognizing that the notion of *missing mass* plays a foundational role. When only a few outputs are sampled from a generative model–such as querying an LLM a handful of times for a prompt–the key question becomes: have we already seen a correct answer, or could the correct output still lie in the part of the distribution we haven't sampled yet? This uncertainty about the correct label remaining unseen—the missing mass—is critical in deciding both whether to keep querying and how much confidence to place in the outputs we have.

To formalize the trade-off between coverage, informativeness, and query cost, we introduce an optimization framework that jointly designs a *query policy* (how many queries to allocate per test point) and a *set map* (how to turn sampled outputs into calibrated prediction sets). Remarkably, both components connect to the classical missing mass problem in statistics (see [15–18]). The optimal query policy corresponds to controlling the *rate of decrease* in missing mass, while the optimal set construction relies on estimating the missing mass itself. We now summarize our main contributions:

**1)** We introduce a novel optimization framework (Section 2) that formally captures the trade-off between coverage, informativeness, and test-time query budget in generative modeling UQ. This reinterprets conformal prediction from a budgeted query perspective and defines two interacting components: the query policy and the prediction set map, which connects sampled outputs to sets.

**2)** We identify two key algorithmic principles that emerge from this framework. First, the optimal query policy prescribes querying each test input until the *rate of decrease in missing mass* falls below a threshold—that is, one should keep querying as long as an additional sample significantly reduces uncertainty. Second, the optimal map from sampled outputs to a set is defined by thresholding a particular conformity score that properly accounts for the *missing mass*. These principles extend conformal prediction to a fundamentally new setting and may be of independent theoretical interest.

**3)** In Section 4, we design a finite-sample algorithm that combines these principles, integrating the estimation of missing mass (and its rate of reduction) into the conformal prediction pipeline while provably maintaining valid, distribution-free coverage guarantees. A key technical contribution is a novel estimator for the rate of decrease in missing mass, derived by revisiting the classical Good–Turing estimator—originally developed to estimate the missing mass itself.

**4)** We show the practical value of our approach through experiments on open-ended LLM tasks. Across three benchmark datasets, we quantify how each algorithmic principle contributes to prediction set informativeness under varying query and coverage constraints. Compared to recent query-based CP baselines [13, 14], our method significantly reduces non-informative sets while maintaining valid coverage guarantees. These highlight the foundational role of our missing mass perspective in CP.

## 1.1 Related works

We briefly discuss closely related works here and defer a full discussion to Appendix A.

**Conformal Language Modeling.** Conformal Language Modeling (CLM) was introduced by [13] and similar ideas further studied by [14, 19–21]. CLM adapts conformal prediction to LLMs by calibrating a set of stopping rules that determine how many outputs to include in a prediction set. However, these methods do not account for uncertainty over unseen generations, and thus only provide valid sets for coverage levels less or equal than the few-shot accuracy of the underlying model. Furthermore, they do not explicitly optimize how the query budget is used across different prompts. In contrast, we provide valid prediction sets for *any* user-defined coverage level and query budget, using a principled framework that bridges conformal prediction with classical missing mass estimation to optimize set informativeness and query efficiency. We also compare against CLM methods in Section 5, demonstrating substantial gains in informativeness, under fixed query budgets.

**Conformal Abstention and Conformal Factuality.** Conformal abstention algorithms refrain from generating a response when uncertainty is high [22–24]. Other works focus on aligning CP with LLM factuality in structured tasks [25–27], or filtering long-form generations by validating sub-claims [28–31]. However, these methods do not construct prediction sets and are thus not directly comparable to ours, though connecting our framework with theirs presents an interesting direction for future work.

## 2 Problem formulation

In this section, we formalize the problem of conformal prediction with a query oracle. Consider a covariate space $\mathcal{X}$ and a potentially infinite label space $\mathcal{Y}$. An input-output pair $(X, Y) \in \mathcal{X} \times \mathcal{Y}$ is drawn from an unknown joint distribution $p(x, y)$, which represents the true data-generating process. For instance, in a text generation task, $X$ could be a prompt and $Y$ the correct or intended response. We assume access to a generative model, referred to as a *query oracle*, which allows us to sample from a conditional distribution $\pi(y \mid x)$. That is, querying the oracle at input $x$ yields an independent sample $y \sim \pi(\cdot \mid x)$. Our goal is to construct prediction sets that provide rigorous coverage guarantees while querying the oracle a finite number of times per test input.

More formally, for a user-specified miscoverage level $\alpha \in (0, 1)$, we seek to ensure the following coverage guarantee:

$$\mathbb{P}_{(X,Y) \sim p}\left[Y \in C(X)\right] \geq 1 - \alpha. \tag{1}$$

Even though $C(X)$ should be constructed using only a finite number of queries to $\pi(y|x)$, which may differ from $p(y|x)$, the coverage constraint in (1) is with respect to $p(y|x)$. This distinction is crucial: while $\pi$ governs the observable behavior of the model, coverage must be guaranteed with respect to the true, unknown distribution $p$.

In classical CP, one defines a nonconformity score function $S : \mathcal{X} \times \mathcal{Y} \to \mathbb{R}$ to measure how atypical a label $y$ is for an input $x$, and constructs prediction sets of the form $C(x) = \{y \in \mathcal{Y} : S(x, y) \leq q\}$, where $q$ is a calibrated threshold. To use such a construction in practice, one must either enumerate the label space $\mathcal{Y}$, as in multi-class classification, or describe the set compactly, such as an interval when $\mathcal{Y} = \mathbb{R}$. However, in tasks such as text generation, sets defined as $\{y \in \mathcal{Y} : S(x, y) \leq q\}$, when $\mathcal{Y}$ is the space of all the text sequences, are not a tractable representation for uncertainty. That is, there is no clear practical way to list all these labels or describe them using an interpretable structure (like an interval). Hence, the standard paradigm of defining a score function and calibrating a threshold

may not fully capture the nature of uncertainty in generative models. Instead, generative models allow for exploring the output space by multiple queries.

What is missing is a perspective that views uncertainty through the lens of querying the generative model–that is, sampling from the oracle. In this view, the information about the true label comes from a finite set of queries: $Z_t(x) = \{y_1^x, \ldots, y_t^x\}$, where $x$ is a test point and $t$ is the number of queries. This multiple-query setting introduces a key limitation: the correct label $Y$ may not be among the queried outputs. This scenario is common in practice–e.g., when using an LLM as the oracle to generate possible responses to a prompt. If none of the generated completions contains the correct answer, we have no signal to recover it. In such cases, there is no choice but to admit high uncertainty and acknowledge that the correct label could lie anywhere in the vast, unseen remainder of $\mathcal{Y}$.

To address this, we introduce a special abstract label EE, short for "Everything Else", which denotes the collection $\mathcal{Y} \setminus Z(x)$. Intuitively, when the model has not yet produced the correct output in its first $t$ queries, the only way to ensure coverage is to include EE in the prediction set. With this formulation, the prediction set $C(x)$ is a subset of $Z(x) \cup$ EE. The CP coverage guarantee $\mathbb{P}(Y \in C(X)) \geq 1 - \alpha$ then admits the interpretation: either the true label $Y$ is among the sampled outputs, or it is captured by EE. Including EE ensures valid coverage even when the true label has not been observed. The key challenge, then, is to avoid including EE unnecessarily—so as to keep prediction sets informative—while still maintaining coverage guarantees across all test points. This creates a fundamental trade-off: querying the oracle more increases the chance of capturing the correct label explicitly, reducing reliance on EE; querying less conserves resources but often necessitates including EE, resulting in less informative predictions. To rigorously navigate this trade-off, we formalize an optimization framework that balances coverage, query cost, and informativeness.

Our framework consists of two components. The first is a **query policy** $T : \mathcal{X} \to \mathbb{N} \cup \{0\}$, which determines *how many* i.i.d. queries to make to the oracle for each input $x$. This effectively allocates the total query budget across test inputs. For each input $x$, we query the oracle $\pi(y|x)$ independently $T(x)$ times, producing a sampled label set $Z(T; x) = \{y_1^x, \ldots, y_{T(x)}^x\}$ for each $x$.

The second component is a **set map** $f : \mathcal{X} \times 2^{\mathcal{Y}} \to 2^{\mathcal{Y}'}$, which converts the queried labels into a prediction set $C(x) = f(x, Z(T; x))$, where $\mathcal{Y}' = \mathcal{Y} \cup$ EE. Given a finite computational query budget $B$ and a user-defined miscoverage rate $\alpha \in [0, 1]$, our goal is to design $T$ and $f$ jointly to ensure valid coverage while maximizing the informativeness of the prediction sets under the budget constraint.

---

**Conformal Prediction with Query Oracle (CPQ)**

$$\underset{f(\cdot),\, T(\cdot)}{\text{minimize}} \quad \mathbb{E}_{X \sim p} \left[ \lambda \mathbb{1}\{\text{EE} \in C(X)\} + \sum_{y \neq \text{EE}} \mathbb{1}\{y \in C(X)\} \right]$$

$$\text{subject to} \quad \Pr_{(X,Y) \sim p}[Y \in C(X)] \geq 1 - \alpha$$

$$\mathbb{E}_{X \sim p}[T(X)] \leq B$$

---

We minimize two forms of uninformative prediction sets: one by the inclusion of EE, the other by the size of the prediction set. Whenever EE $\in C(x)$, the conditional coverage at $x$ is trivially satisfied: $\mathbb{P}[Y \in C(x) \mid X = x] = 1$. Thus, including EE guarantees coverage but offers no useful information. Penalizing EE is therefore essential: the challenge lies not in achieving coverage, but in doing so while using EE as infrequently as possible. The parameter $\lambda \geq 0$ controls the penalty ratio. We focus on the regime where $\lambda \gg 1$, expressing a strong preference for minimizing the use of EE across the population. However, the second term remains important to prioritize smaller sets among those that avoid EE maximally. In the next section, we analyze this objective to uncover two key algorithmic principles. These principles will ultimately guide the design of our practical, finite-sample algorithm.

## 3 Algorithmic Principles

The CPQ problem introduced above is a joint optimization over two components: the query policy $T(\cdot)$ and set map $f(\cdot)$. In this section, we adopt a decoupled analysis, splitting the problem into two stages. First, we fix a query budget and ask: *What is the optimal query policy for allocating queries to minimize the chance of missing the correct label?* Then, given a fixed query policy, we ask: *What is the optimal set map for constructing informative prediction sets while ensuring valid coverage?*

It is worth noting that this decoupled analysis only approximates the full CPQ solution, as it breaks the joint optimization over $T$ and $f$. Accordingly, optimality in this section refers to the best solution within each stage, rather than the overall joint optimum.

To answer these questions, we work in the population regime, assuming the query oracle $\pi(y \mid x)$ is the same as the true conditional distribution $p(y \mid x)$. *Consequently, we assume throughout this section that $\pi \equiv p$; i.e., the query oracle is perfect.* This idealized setting allows us to derive two algorithmic principles—one for query policy and one for prediction set construction—that form the foundation of our finite-sample algorithm. In Section 4, we show how to apply these principles with any black-box query oracle (e.g., an LLM), particularly when $\pi(y|x) \neq p(y|x)$, to construct practical prediction sets with finite-sample coverage guarantees. Proofs are deferred to Appendix B.

## 3.1 Principle 1: Optimal Querying Policy by Missing Mass Minimization

We now focus on the query policy, aiming to allocate queries across covariate points to minimize the chance of missing the correct label. If computational resources were unlimited, we could query the oracle exhaustively for each input $x$, fully uncovering the label distribution and removing the need for the abstract label EE. But under a finite budget, we must query strategically—balancing where and how much to query—an objective naturally captured by the concept of *missing mass*.

Formally, the *missing mass* for a covariate $x$ after $t$ queries is defined as the probability that the true label $Y$ is not among the sampled set $Z_t(x)$:

$$\theta(x, t) = \Pr_{Y, Z_t(x)} \big[ Y \notin Z_t(x) \mid X = x \big],$$

where $Z_t(x)$ consists of $t$ i.i.d. samples from $p(y \mid x)$. Intuitively, $\theta(x, t)$ measures residual uncertainty–the chance that $t$ independent draws from $p(y \mid x)$ fail to capture the true label. As $t$ increases, $\theta(x, t)$ naturally decreases, and does so with *diminishing returns*: each additional query is less likely to reduce uncertainty than the previous one. To make this precise, define the finite difference as $\Delta(x, t) := \theta(x, t+1) - \theta(x, t)$. For each $x$, $\Delta(x, t)$ is negative and non-decreasing in $t$, meaning $\theta(x, t)$ is non-increasing with diminishing returns (see Appendix B for proofs).

These properties make missing mass a natural objective for query policy. For each input $x$, increasing the number of queries $t$ reduces the probability of missing the true label—i.e., lowers $\theta(x, t)$—and eventually, we may no longer need to include EE in the prediction set for that $x$. However, since our total query budget is limited, we cannot afford to exhaustively query all inputs. This raises the core question: how should we allocate our finite budget across different covariates to minimize overall uncertainty? That is, which inputs should receive more queries to reduce reliance on EE most effectively? This naturally leads to the following optimization problem:

$$
\begin{aligned}
\min_{T(\cdot):\mathcal{X} \to \mathbb{N} \cup \{0\}} \quad & \mathbb{E}_X\big[\theta(X, T(X))\big] \\
\text{subject to} \quad & \mathbb{E}_X[T(X)] \leq B.
\end{aligned}
\tag{2}
$$

**Theorem 3.1** (Optimal Query Policy). *Assuming $X$ is a continuous random variable, let $T^*(\cdot)$ be the optimal solution to the optimization problem (2). Then, there exists a constant $\beta^* \in \mathbb{R}$ such that, for all $x \in \mathcal{X}$ almost surely, the optimal query size $T^*(x)$ satisfies:*

$$\Delta(x, T^*(x) - 1) \leq \beta^* < \Delta(x, T^*(x) + 1) \tag{3}$$

This condition implies that at the optimal query number $T^*(x)$ for each $x$, the discrete derivative $\Delta(x, T^*(x))$ from one more query is approximately equal to the threshold $\beta^*$ (note that $\Delta(x, t)$ is non-decreasing in $t$). This result suggests a simple and intuitive principle: continue querying the oracle for a given $x$ as long as doing so substantially reduces the missing mass. In other words, we should stop sampling when the gain from an additional query falls below a threshold $\beta^*$. This behavior is directly driven by the diminishing returns property of $\theta(x, t)$ and constitutes our first algorithmic principle. This insight guides the query policy in our finite-sample algorithm in Section 4, where we replace the exact derivative $\Delta(x, t)$ with a data-driven estimate $\hat{\Delta}(x, t)$, and stop querying when $\hat{\Delta}(x, t) \leq \beta^*$, with $\beta^*$ calibrated from finite samples to satisfy the query budget $B$.

### 3.2 Principle 2: Optimal Prediction Sets by Missing Mass Estimation

In this section, we assume we are given access to a predetermined and known query policy function $T : \mathcal{X} \to \mathbb{N}$, which specifies the number of i.i.d. queries made to the oracle for each input $x$. For each $x \in \mathcal{X}$, we denote the resulting set of sampled labels by $Z(T, x) = \{y_1^x, \ldots, y_{T(x)}^x\}$. With these samples in hand, our goal is to construct prediction sets that satisfy the desired coverage guarantee while being as informative as possible.

To achieve this, we formulate an optimization problem to determine the best possible prediction sets under coverage constraints. The primary goal is twofold: (1) minimize the inclusion of the abstract label EE, as its presence indicates complete uncertainty, and (2) among sets with minimal inclusion of EE, minimizing the prediction set sizes. Reminding $f : \mathcal{X} \times 2^{\mathcal{Y}} \to 2^{\mathcal{Y}'}$ and $C(x) = f(x, Z(T; x))$ from Section 2, we introduce:

$$
\begin{aligned}
\min_{f(\cdot)} \quad & \mathbb{E}_X \left[ \lambda \mathbb{1}\{\text{EE} \in C(X)\} + \sum_{y \neq \text{EE}} \mathbb{1}\{y \in C(X)\} \right] \\
\text{subject to} \quad & \Pr_{X,Y} \left[ Y \in C(X) \right] \geq 1 - \alpha,
\end{aligned}
\tag{4}
$$

The parameter $\lambda \geq 0$ balances the trade-off between avoiding EE and keeping prediction sets small. We are particularly interested in the regime where $\lambda$ is large. This reflects a strict preference for minimizing the use of EE, while still allowing the optimization to differentiate among prediction sets that achieve the same frequency of EE inclusion. The inclusion of the second term ensures that among all valid prediction rules minimizing EE, we favor the most informative ones with smaller set sizes. Next, we characterize the structure of the optimal set map solution to (4) in the following theorem.

**Theorem 3.2** (Optimal Set-Assignment Policy). *Assuming $X$ is a continuous random variable, for sufficiently large values of $\lambda$, the optimal solution $f_\lambda^*$ to the optimization problem (4) has the following structure: there exists a scalar threshold $q^* \in \mathbb{R}^+$ satisfying*

$$
f^*(x, Z(x)) = \{y \in Z(x) \cup \{EE\} : S(x, y) \leq q^*\}, \quad \textit{almost surely for every } x.
$$

*Also, defining $p(\text{EE}|x) = \Pr_Y \left[ Y \notin Z_t(x) \mid X = x \right]$, we have,*

$$
S(x, y) = \begin{cases} 1 - p(y \mid x), & \textit{if } y \neq \textit{EE}, \\ 2 - p(y \mid x), & \textit{if } y = \textit{EE}. \end{cases}
\tag{5}
$$

Theorem 3.2 shows that the optimal prediction sets can be constructed by thresholding a conformity score $S(x, y)$. This score prioritizes explicitly sampled labels over the abstract label EE, ensuring that EE is included only if necessary. Specifically, EE is assigned a score of $2 - p(\text{EE} \mid x)$, where $p(\text{EE} \mid x)$ corresponds exactly to the missing mass. This means EE is most likely to be included when the missing mass is high–an intuitive and desirable behavior. Moreover, this result generalizes the classic finding in conformal prediction that optimal prediction sets minimizing size under a coverage constraint are obtained by thresholding $1 - p(y \mid x)$, in classification and regression [32, 33].

To summarize, we have derived two foundational principles: one connecting the optimal query policy to the derivative of the missing mass, and the other connecting the optimal set map to the missing mass itself through an optimal conformity score. In the next section, we build upon these principles to design a practical finite-sample algorithm.

## 4 Finite Sample Algorithm

In this section, we present our finite-sample algorithm, which consists of two modules, each carefully built upon the algorithmic principles derived in Section 3. The query module relies on an estimator of the *missing mass derivative*, denoted $\hat{\Delta}(x, t)$, while the calibration module uses an estimator of the *missing mass* itself, $\hat{\theta}(x, t)$–both of which we detail below.

**Estimating Missing Mass and Its Derivative.** Let $\mathcal{Y}$ be the label space, and suppose we observe a sequence of $t$ i.i.d samples $Z_t(x) = \{y_1^x, \ldots, y_t^x\} \sim \pi(y|x)$, i.e., samples from the oracle. The

missing mass, $\theta(x, t)$, is defined as the total probability of all labels in $\mathcal{Y}$ that have not been observed in the sample $Z_t(x)$. For each integer $r \geq 0$, let $N_r(x, t)$ denote the number of distinct labels that occur exactly $r$ times in the sample $Z_t(x)$. In other words, $N_r(x, t) = |\{y \in Z_t(x) : \#(y) = r\}|$, where $\#(y)$ denotes the number of times the label $y$ appears in the sample $Z_t(x)$.

The classical Good-Turing estimator approximates the missing mass based on the labels seen exactly once, aka **singletons**. The intuition is simple in that if many labels appear only once, it is likely that there are more yet-unseen labels with comparable probabilities. This yields the estimator $\boxed{\hat{\theta}(x, t) := \frac{N_1}{t}}$. In fact, Good-Turing estimators also provide estimates for seen labels. For $y \in Z_t(x)$, we estimate $p(y \mid x)$ using the Good–Turing formula: $\hat{\omega}(y \mid x) = \frac{r+1}{t} \cdot \frac{N_{r+1}}{N_r}$, where $r$ is the number of times $y$ appears in $Z_t(x)$. Hence, we estimate the conformity score derived in our optimal set construction (see Eq. (5)) by $\hat{S}(x, y) = \begin{cases} 1 - \hat{\omega}(y \mid x), & \text{if } y \in Z_t(x) \\ 2 - \hat{\theta}(x, t), & \text{if } y \notin Z_t(x) \end{cases}$.

On the other hand, the query module requires an estimate for the *missing mass derivative* $\Delta(x, t) = \theta(x, t + 1) - \theta(x, t)$, which captures the reduction in missing mass from drawing an additional sample. By revisiting the original calculations behind the classical Good-Turing estimator, we derive the following novel estimator for the derivative: $\boxed{\hat{\Delta}(x, t) := -\frac{2N_2}{t^2}}$.

Interestingly, we see that while the Good-Turing estimator relates the missing mass to the count of *singletons*, our estimator for the derivative reveals that the count of **doubletons**, number of unique labels that appear twice, is a good proxy for the rate at which the missing mass decreases. A detailed derivation is provided in Appendix D.1. Furthermore, we will showcase the empirical performance of this estimator on two synthetic distributions in Appendix D.2, along with comparisons against a natural baseline: the plug-in estimate of the derivative computed by taking finite differences of the Good-Turing missing mass estimates at successive values of $t$.

**Algorithm.** Assume we have access to a query oracle $\pi(y|x)$ that approximates–but may not perfectly match–the true conditional distribution $p(y|x)$. By querying this oracle, we can draw independent samples from $\pi(y|x)$ for each input $x$, and compute quantities such as the missing mass (or its derivative) as needed. Additionally, we are given calibration data $\mathcal{D}_{\text{cal}} = (X_i, Y_i)_{i=1}^N$ drawn from the ground truth distribution $p(x, y)$, as is standard in CP.

To tune the query threshold $\beta^*$, we first partition the calibration data $D_{\text{cal}}$ into two disjoint subsets $D_{\text{cal}_1}$ and $D_{\text{cal}_2}$. The first subset $D_{\text{cal}_1}$ is used exclusively for tuning $\beta^*$ as follows: for each input $x \in D_{\text{cal}_1}$, draw a set of queries $y_{1:T(x)} \sim \pi(y|x)$, where $T(x)$ is the smallest integer number at which $\hat{\Delta}(x, T(x)) \leq \beta^*$. Given a query budget constraint $B$, select $\beta^*$ such that the average number of queries $\frac{1}{|D_{\text{cal}_1}|} \sum_x T(x) \leq B$. Since $\beta^*$ is a scalar, this can be done via exhaustive search on a grid of values. Once $\beta^*$ is fixed, we apply our algorithm presented in Algorithm 1.

---

**Algorithm 1** Conformal Prediction with Query Oracle (CPQ)

---

**Input:** Query oracle $\pi(y \mid x)$, conformity score $\hat{S}(x, y)$, calibration data $\mathcal{D}_{\text{cal}_2}$, test point $x_{\text{test}}$, miscoverage $\alpha$, query budget $B$, missing-mass estimator $\hat{\Delta}(x, t)$, threshold $\beta^*$

> Query Module → Principle 1
>
> - For each $x \in \mathcal{D}_{\text{cal}_2} \cup \{x_{\text{test}}\}$:
>   - Sample $y_{1:T(x)} \sim \pi(y \mid x)$ until $\hat{\Delta}(x, T(x)) \leq \beta^*$. Let $Z(x) = \{y_1, \ldots, y_{T(x)}\}$.

> Calibration Module → Principle 2
>
> - For each $(x_i, y_i) \in \mathcal{D}_{\text{cal}_2}$ compute $s_i = \hat{S}(x_i, y_i)$.
> - Set $q^* = \text{Quantile}_{1-\alpha}\big(s_1, \ldots, s_{|\mathcal{D}_{\text{cal}_2}|}, \infty\big)$.

**Output:** $C(x_{\text{test}}) = \big\{ y \in Z(x_{\text{test}}) \cup \{\texttt{EE}\} : \hat{S}(x_{\text{test}}, y) \leq q^* \big\}$.

---

The algorithm consists of two stages, each directly motivated by the algorithmic principles derived in Section 3. In the first stage-the *query module*-we determine how many queries to draw for each input $x$. We query sequentially from the query oracle $\pi(y \mid x)$ (e.g. an LLM), one at a time, and after each draw, we update the estimated missing mass derivative $\hat{\Delta}(x, t)$. Guided by **principle 1**, we continue sampling until $\hat{\Delta}(x, t)$ falls below the threshold $\beta^*$. The result of this stage is a set $Z(x)$ of observed labels for each input $x$, along with the associated estimated missing mass for the fallback label EE. In the second stage- the *calibration module*-motivated by **principle 2**, we calculate the conformity scores $\hat{S}(x, y)$ on a held-out calibration set $D_{cal_2}$ as described earlier in this Section. We then compute the $(1 - \alpha)$-quantile of the conformity scores on $D_{cal_2}$ (adjusted with $\infty$ for proper debiasing), and use this threshold to construct prediction sets for test inputs, following the standard split conformal procedure.

The following theorem guarantees the distribution-free coverage validity of our algorithm.

**Theorem 4.1** (Coverage Validity). *Assuming $\mathcal{D}_{test}$ and $\mathcal{D}_{cal_2}$ are exchangeable, we have:*
$$\Pr[Y_{test} \in C(X_{test})] \geq 1 - \alpha,$$
*where the probability is over $(X_{test}, Y_{test})$ and $\mathcal{D}_{cal_2}$.*

In summary, CPQ adaptively query the oracle guided by an estimation of derivative of the missing mass, and then make prediction sets guided by Good-Turing estimate of the missing mass itself.

## 5   Experimental Results

We begin by outlining our experimental setup, then present empirical evaluations along two main axes: (i) a component-wise analysis isolating the impact of optimal querying and optimal conformal calibration (Section 3), and (ii) a comparison against state-of-the-art conformal language modeling baselines, including CLM [13] and its recent variant, SCOPE-Gen [14].

**Datsets and Models**. We evaluate on three benchmark datasets using two leading LLMs, adapting all tasks to open-eneded generation by removing any multiple-choice structure. Generations are lexically normalized and marked correct only if they exactly match the ground truth answer; i.e evaluating using the *exact match* metric. The datasets are: (i) *BBH Geometric Shapes [34] (250 prompts)*: Visual reasoning from SVG paths, with responses generated using LLaMA-3 8B-Instruct [35]. (ii) *GSM8K [36] (300 randomly selected prompts)*: Multi-step arithmetic reasoning, answers from Mixtral-8x7B-Instruct [37]. (iii) *BBH Date Understanding [34] (250 prompts)*: Temporal reasoning; responses generated using LLaMA-3 8B-Instruct.

**Evaluation Metrics**. Our goal is to construct prediction sets that are both *valid* and *informative*. We report three key evaluation metrics. First, ***Empirical Coverage***: the fraction of test examples whose prediction set contains either the correct answer or EE, either ensures validity ( see Section 2). Second, *EE **fraction*** measures how often EE appears; lower fractions indicate the model more often explicitly captures the correct answer without relying on fallback coverage via EE. Third, ***Average set size***: the average number of *seen* labels per prediction set. While larger sets generally imply less informative sets, a larger set without EE conveys more information than a smaller set with EE, as the former expresses uncertainty within observed outputs, whereas the latter signals residual uncertainty over the entire unobserved label space. Together, these metrics capture the tradeoff between coverage and informativeness. An ideal prediction set achieves target coverage with minimal reliance on EE.

**Clustering**. Clustering is a key step in our pipeline. Since LLMs produce lexically varied outputs that convey the same meaning, we group generations into *semantic equivalence* classes (*clusters*), each corresponding to a single label $y \in \mathcal{Y}$. We use LLaMA-3-8B-Instruct model to decide if two generations semantically equivalant and assign them to the same cluster if so. This approach has proven effective for handling complex and unstructured outputs [19, 38]. Prompts and implementation details are provided in Appendix C.5. Each cluster's frequency is used to estimate the *missing mass* (probability of *unseen* clusters) and its *derivative* (see Section 4). Probabilities for *seen* clusters are computed by normalizing frequencies and scaling to form a valid distribution over both seen and unseen clusters. Importantly, our finite sample algorithm is modular: it works with any clustering or probability estimation method. As long as clustering and associated probabilities are well defined and valid, our method applies.

**Calibration and sampling procedures.** For each dataset, we randomly split examples equally into calibration and test sets. On the calibration set, we tune CPQ's sampling threshold $\beta^*$ to meet the

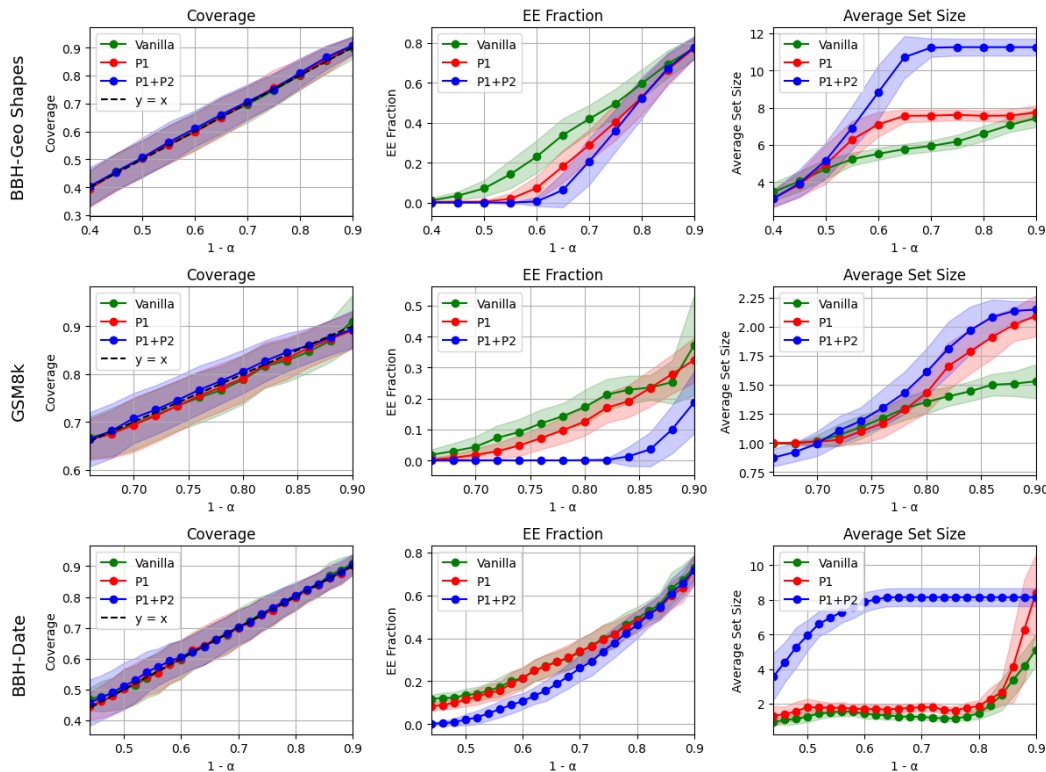

Figure 1: Performance of the three algorithmic variants (Vanilla, P1, P1+P2 : corresponds to our full finite sample algorithm, i.e CPQ) across Geo Shapes ($B = 30$), GSM8k ($B = 7$), and BBH-Date ($B = 20$). Each row shows coverage, EE fraction, and average set size as a function of $1 - \alpha$.

target average query budget and estimate the threshold $q^*$ for constructing prediction sets. All results are averaged over 50 random splits of calibration and test data.

## 5.1 Fine-grained Component-wise Analysis

To assess contributions of each algorithmic principle (Section 3), we compare three progressively refined variants: (i) **Vanilla**: A baseline with a fixed, non-adaptive querying strategy—the same number of generations per input—and a simple, yet valid calibration rule. While not optimal, this baseline serves as a reasonable starting point. Calibration details are provided in the Appendix C.1. (ii) **Principle 1**: Adds our adaptive querying, adjusting the number of queries based on the estimated missing mass derivative, with calibration unchanged. (iii) **Principle 1 + 2**: Combines both optimal querying and conformal calibration, representing the full CPQ algorithm in Section 4.

Figure 1 shows results on all benchmark datasets. We observe consistent gains from incorporating each algorithmic principle, with the full CPQ algorithm (both principles combined) achieving the largest reduction in the fraction of prediction sets that include the fallback label EE, while maintaining valid coverage. The query budget $B$ is fixed per dataset, while the coverage level $1 - \alpha$ is varied. Budgets were chosen to reflect reasonable intermediate values based on the few-shot model accuracy for each dataset. Additional results across a range of budgets can be found in Appendix C.2.

We see that CPQ navigates a clear trade-off between reducing reliance on the fallback label EE and using observed labels. Both Principle 1 and the full CPQ algorithm (Principles 1+2) actively leverage the queried labels to avoid including EE while maintaining valid coverage. The additional reduction in EE fraction achieved by the full CPQ algorithm naturally comes at a cost: the number of non-EE labels included in the prediction set increases, leading to larger set sizes. This behavior is expected since aggressively minimizing reliance on EE requires exploiting the observed labels more fully to preserve coverage.

| Dataset | Algorithm | Nom. Cov. | Emp. Cov. | EE Frac. |
|---------|-----------|-----------|-----------|----------|
| **Geo** | CLM | 0.60 | $0.58 \pm 0.038$ | $0.40 \pm 0.047$ |
| | Scope-Gen | 0.60 | $0.68 \pm 0.080$ | $0.38 \pm 0.22$ |
| | CPQ | 0.60 | $0.61 \pm 0.06$ | $\mathbf{0.07} \pm 0.07$ |
| **GSM8K** | CLM | 0.95 | $0.93 \pm 0.03$ | $0.70 \pm 0.11$ |
| | Scope-Gen | 0.95 | $0.93 \pm 0.05$ | $0.61 \pm 0.26$ |
| | CPQ | 0.95 | $0.95 \pm 0.02$ | $\mathbf{0.16} \pm 0.14$ |
| **Date** | CLM | 0.70 | $0.68 \pm 0.07$ | $0.32 \pm 0.11$ |
| | Scope-Gen | 0.70 | $0.78 \pm 0.07$ | $0.51 \pm 0.11$ |
| | CPQ | 0.70 | $0.71 \pm 0.07$ | $\mathbf{0.25} \pm 0.08$ |

Table 1: Comparison of nominal coverage, empirical coverage, and the fraction of sets that contain the fallback label (EE) across three benchmark datasets (Geometric shapes (Geo), GSM8K, Date understanding (Date)) for three methods: CLM, Scope-Gen, and our proposed CPQ.

## 5.2 Comparison with Conformal Baselines

We now compare **CPQ** to two recent conformal prediction methods for large language models: **CLM** [13] and its variant **SCOPE-Gen** [14]. While both represent state-of-the-art in this space, they are not out-of-the-box comparable with CPQ in two key ways. First, neither accounts for the missing mass—the residual probability over unseen labels represented by EE in our framework. As a result, they may fail to provide valid configurations at higher coverage levels, especially when the correct answer isn't among the sampled outputs. Second, they lack an explicit mechanism to control query budget: the number of model queries varies across coverage levels and is not directly tunable.

To enable a meaningful comparison, we evaluate **CLM** and **SCOPE-Gen** using their original procedures, with one adjustment: we augment their output space to include the abstract label EE alongside sampled responses. The underlying logic and mechanisms remain unchanged; we simply extend the prediction space to reflect the possibility of unseen correct label, which is necessary for a complete coverage analysis. This enables us to assess how often these baselines would have needed to include EE to satisfy coverage validity. Since, there is no principled way to configure these baselines to target a specific budget, we first measure their average query usage. We then tune **CPQ**'s querying threshold $\beta^*$ to match this budget. All methods are thus evaluated on equal footing at the same nominal coverage level and under the same average query budget.

As shown in Table 1, CPQ dramatically reduces reliance on EE. For example, on GSM8k at $95\%$ nominal coverage, CPQ achieves the desired coverage with an EE fraction of $16.5\%$, versus $70.4\%$ for CLM and $61\%$ for SCOPE-Gen under the same budget constraints. Moreover, CPQ not only offers more informative prediction sets but also maintains tighter coverage, especially in high-coverage regimes where baselines struggle.

## 6 Conclusion and Limitations

We presented a principled framework for UQ by introducing a novel missing mass perspective. we derived two algorithmic principles that guide optimal query policy and prediction set construction. Our finite-sample algorithm integrates these insights and yields significantly more informative prediction sets compared to existing conformal methods for LLM UQ. Our method relies on estimation of missing mass and its derivative, which can be challenging in very low query regimes.

## 7 acknowledgments

This work was supported by the NSF Institute for CORE Emerging Methods in Data Science (EnCORE) and the ASSET (AI-Enabled Systems: Safe, Explainable and Trustworthy) Center.

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

**Table of Contents**

# A   Extended Related Works

**Conformal Prediction.** The notion of prediction sets originates from classical work on tolerance regions in statistics [39, 40]. However, the modern formulation of Conformal Prediction (CP), which provides distribution-free, finite-sample validity guarantees, was introduced by [5, 7, 41]. Since then, CP has emerged as a standard framework for uncertainty quantification, particularly in classification [11, 42, 43] and regression tasks [44–46]. A growing body of work have then focused on improving the set size (length) efficiency of conformal prediction sets [28, 32, 33, 47, 48]. These developments reflect the increasing demand for flexible and reliable uncertainty quantification in modern predictive systems.

**Conformal Prediction for LLMs.** Recent work has explored conformal prediction as a principled tool for uncertainty quantification in Large Language Models (LLMs), where outputs are open-ended and unbounded. Conformal Language Modeling [13] introduced a sampling-and-filtering approach that generates candidate responses until a calibrated stopping rule guarantees, with high probability, that at least one correct answer lies in the set. Generative Prediction Sets (GPS) [20] recasts the problem as conformal regression on the number of samples required for a correct output, using the resulting distribution to infer minimal draw count needed to achieve nominal coverage. SCOPE-Gen [14] proposes a sequential pruning strategy using greedy admissibility filters, leveraging a Markov factorization to reduce verification costs during calibration. APIisEnough [21] offers a black-box approach that defines nonconformity via sampling frequencies and semantic similarity; their approach can be integrated in our modular framework seamlessly.

Several complementary directions have further adapted CP to the generative language setting: token-level CP for non-exchangeable generation [25], representation-level conformal alignment, filtering methods for long-form factuality guarantees [28, 29, 31], multi-group uncertainty quantification in structured text [30], and CP for enumerable, discrete output spaces such as multiple-choice tasks [27]. While all these methods offer valid coverage, they vary in efficiency, granularity, and scope, and none explicitly incorporate missing mass estimation as a means to reason about unseen correct responses to capture the full output space. Moreover, they do not account for or optimize under an explicit query budget, a central component of our framework. In contrast, out method address both dimensions-coverage in the presence of unobserved labels and efficient query allocation-through a unified, theoretically grounded approach.

**Conformal abstention for LLMs.** An alternative to constructing prediction sets is to enable selective prediction: allowing the LLM to abstain from responding when uncertain. This line of work aims to mitigate erroneous outputs by identifying in puts where the model's predictions are unreliable. In particular, [22] apply conformal risk control to bound the probability of hallucination and derive abstention rules that trigger whenever the estimated risk exceeds a calibrated threshold.Moreover, [23] integrate CP with reinforcement learning to learn abstention policies that adaptively respond to task difficulty and distributional shifts. Separately, [24] introduce an information-theoretic decomposition of uncertainty into epistemic and aleatoric components, leveraging the epistemic signal to guide abstention decisions.

While these methods share the goal of reliable decision-making under uncertainty in LLMs, they differ from our approach in that they do not produce explicit prediction sets, and therefore cannot be directly compares. One could, in principle, adapt intermediate quantities from our method-such as prediction set size or estimated missing mass-as abstention criteria, which can be an interesting venue for future work.

**Broader Uncertainty Quantification for LLMs.** Our work is informed by a broad literature on uncertain quantification (UQ) for LLMs that extends beyond conformal prediction. A substantial body of research focuses on mitigating hallucinations in LLM outputs, employing techniques ranging from direct uncertainty estimation [4, 49–51] to strategies that generate multiple responses to probe and analyze the output space [52]. Prior research has observed that semantic disagreement among sampled responses correlates with hallucinations risk, motivating a suite of detection methods based on self-consistency, token-level log-probability, or verifier-based models [30, 38, 53]. While these heuristics have demonstrated empirical success, they generally lack formal coverage guarantees and often require extensive sampling or auxiliary models.

**Missing Mass.** The missing mass problem- estimating the total probability of outcomes not observed in a given sample- has been extensively studied under the assumption of independent and identically

distribution (i.i.d) data. Theoretical results have established concentration inequalities for the missing mass around its expectation [54–57], studying the stability and predictability of this quantity in large-sample regimes. Central to practical estimation, the classical Good-Turing (GT) estimator, first introduced by [58], has been analyzed extensively, with multiple variants developed to improve its finite-sample performance [59–64]. Confidence intervals for missing mass were obtained using the GT estimator in [65] and subsequently refined by [66]. Building upon these ideas, [67] developed the "Good-Toulmin" estimator, extending the missing mass framework to the species-discovery problem. Though conceptually related, species discovery-estimating how many new previously unseen categories are expected to appear in an enlarged sample-and missing mass estimation-which quantifies unseen probability mass-are fundamentally different in objective and interpretation.

# B  Proofs

## B.1  Proof of Theorem 3.1

We first start by reviewing the theorem statement. Let $\theta(x,t)$ be the missing–mass curve defined in Section 3.1, and $\Delta(x,t) = \theta(x,t+1) - \theta(x,t)$. There exists a threshold $\beta^* \leq 0$ such that, almost surely,
$$\Delta\big(x, T^*(x) - 1\big) \ \leq \ \beta^* \ < \ \Delta\big(x, T^*(x) + 1\big),$$
or, $T^*(x) = 0$ whenever $\Delta(x,0) \leq \beta^*$.

Let $\mathcal{T} := \{T : \mathcal{X} \to \mathbb{N}_{\geq 0} \text{ measurable} \mid \mathbb{E}[T(X)] \leq B\}$ and let $T^* \in \mathcal{T}$ be an optimal solution.

For $\beta \leq 0$ define the measurable sets
$$A_\beta := \{x : \Delta(x, T^*(x) - 1) > \beta, \text{and } T^*(x) > 0\}, \qquad B_\beta := \{x : \Delta(x, T^*(x) + 1) \leq \beta\}.$$

Because $\Delta(x, T^*(x)) \leq \Delta(x, T^*(x) + 1)$, the sets $A_\beta$ and $B_\beta$ are disjoint. We can now prove the following claim.

*Claim.* $p(A_\beta)\,p(B_\beta) = 0$ for every $\beta \leq 0$.

*Proof of the claim.* Assume $p(A_\beta), p(B_\beta) > 0$. Take measurable $A \subseteq A_\beta$, $B \subseteq B_\beta$ with $p(A) = p(B) = \eta > 0$ (this exists due to the assumption that $X$ is a continuous random variable) and set
$$T'(x) := \begin{cases} T^*(x) - 1, & x \in A_\beta, \\ T^*(x) + 1, & x \in B_\beta, \\ T^*(x), & \text{otherwise.} \end{cases}$$

Then we have,
$$\mathbb{E}[T'(X)] = \mathbb{E}_X\big[T^*(X) - \mathbf{1}[X \in A_\beta] + \mathbf{1}[X \in B_\beta]\big] = \mathbb{E}[T^*(X)] \leq B,$$

therefore, $T' \in \mathcal{T}$. Furthermore,
$$\begin{aligned}
\mathbb{E}\big[\theta(X, T'(X)) - \theta(X, T^*(X))\big] & \\
&\stackrel{(a)}{=} -\mathbb{E}[\mathbf{1}[X \in A_\beta]\,\Delta(X, T^*(X) - 1)] + \mathbb{E}[\mathbf{1}[X \in B_\beta]\,\Delta(X, T^*(X))] \\
&\stackrel{(b)}{\leq} -\mathbb{E}[\mathbf{1}[X \in A_\beta]\,\Delta(X, T^*(X) - 1)] + \mathbb{E}[\mathbf{1}[X \in B_\beta]\,\Delta(X, T^*(X) + 1)] \\
&\stackrel{(c)}{<} \mathbb{E}[\mathbf{1}[X \in A_\beta]\,\beta] + \mathbb{E}[\mathbf{1}[X \in B_\beta]\,\beta] \\
&\stackrel{(d)}{=} -\eta\beta + \eta\beta = 0,
\end{aligned}$$

where (a) follows from the definition of $T'$, (b) stems from Lemma B.1 which indicates the diminishing return property, (c) follows from the definitions of $A_\beta$ and $B_\beta$, and finally, (d) is due to the definition of $\eta$. This is a contradiction with the optimality of $T^*$, hence we proved the claim.

**Existence and characterization of the threshold $\beta^*$.** Define the threshold $\beta^*$ by setting
$$\beta^* \ := \ \inf\big\{\beta \leq 0 : p(A_\beta) = 0\big\}.$$

Intuitively, this threshold separates covariate points into two groups: those for which an additional query would yield a marginal improvement strictly greater than $\beta^*$, and those for which the marginal improvement from additional queries is at most $\beta^*$. To see why $\beta^*$ is indeed the correct threshold, suppose there existed covariates violating the threshold condition at this $\beta^*$. Then, we could slightly perturb the threshold, obtaining a nearby threshold $\beta'$ such that both sets $A_{\beta'}$ and $B_{\beta'}$ simultaneously have positive probability. But this situation would directly contradict the claim we proved earlier, which ensures that at no threshold can both $A_\beta$ and $B_\beta$ have positive probability. Thus, no violation at threshold $\beta^*$ can occur, confirming that $\beta^*$ is precisely the desired threshold.

We now formalize this intuition precisely. Define the violation probabilities

$$f(\beta) := p(A_\beta) \quad \text{and} \quad g(\beta) := p(B_\beta), \quad \beta \le 0.$$

Observe that enlarging the threshold $\beta$ reduces the set $A_\beta$ and expands the set $B_\beta$. Therefore, the function $f(\beta)$ is non-increasing and right-continuous, and $g(\beta)$ is non-decreasing and left-continuous. Additionally, at $\beta = 0$, we have $f(0) = 0$, since by construction $\Delta(x,t) \le 0$.

By right-continuity of $f(\cdot)$, it follows immediately from the definition of $\beta^*$ that

$$p(A_{\beta^*}) = f(\beta^*) = 0.$$

Next, assume towards contradiction that $p(B_{\beta^*}) > 0$. By left-continuity of $g(\cdot)$, there would exist an $\varepsilon > 0$ sufficiently small so that $p(B_{\beta^* - \varepsilon}) > 0$. However, by the definition of $\beta^*$, lowering the threshold to $\beta^* - \varepsilon$ would yield $p(A_{\beta^* - \varepsilon}) > 0$. Thus, at threshold $\beta^* - \varepsilon$, both $A_{\beta^* - \varepsilon}$ and $B_{\beta^* - \varepsilon}$ would simultaneously have positive probability, contradicting the claim we previously established. Hence, we must have

$$p(B_{\beta^*}) = 0.$$

Finally, since $p(A_{\beta^*}) = 0$ and $p(B_{\beta^*}) = 0$, we have for almost every $x$:

$$\Delta\big(x, T^*(x) - 1\big) \le \beta^* < \Delta\big(x, T^*(x) + 1\big).$$

In the corner case where $\Delta(x,0) \le \beta^*$, the definition of $A_{\beta^*}$ forces the optimal query count $T^*(x) = 0$. This establishes precisely the threshold characterization asserted in the theorem, thereby completing the proof. $\qquad \square$

We now prove the following lemma, which we used in the above proof.

**Lemma B.1** (Diminishing Returns). *For every fixed covariate $x \in \mathcal{X}$, the marginal*

$$\Delta(x,t) = \theta(x, t+1) - \theta(x,t), \qquad t \ge 0,$$

*is strictly negative and non-decreasing in $t$; that is,*

$$\Delta(x,t) < 0 \quad \text{and} \quad \Delta(x, t+1) \ge \Delta(x,t) \quad \forall t \ge 0.$$

Lemma B.1 establishes that as $t$ increases, the missing mass $\theta(x,t)$ naturally decreases, and does so with diminishing returns, meaning each additional query is less likely to reduce the uncertainty than the previous one. Thus the derivative of the missing mass, namely $\Delta(x,t)$ is negative and non-decreasing in $t$.

*Proof.* The missing mass is

$$\theta(x,t) = \Pr_{Y, Z_t(x)}[Y \notin Z_t(x) \mid X = x] = \mathbb{E}_{Y, Z_t(x) \mid X = x}[\mathbf{1}\{Y \notin Z_t(x)\}].$$

Applying law of total expectation

$$\theta(x,t) = \mathbb{E}_{Y \mid X = x} \, \mathbb{E}_{Z_t(x) \mid Y, X = x}[\mathbf{1}\{Y \notin Z_t(x)\}].$$

and evaluating the inner expectation Conditioned on $Y = y$, the $t$ draws in $Z_t(x)$ miss $y$ with probability $(1 - p(y \mid x))^t$, hence

$$\theta(x,t) = \mathbb{E}_{Y \mid X = x}\big[(1 - p(Y \mid x))^t\big].$$

The, the finite difference becomes:

$$\Delta(x,t) = \theta(x, t+1) - \theta(x, t)$$
$$= \mathbb{E}_Y\big[(1 - p(Y \mid x))^{t+1} - (1 - p(Y \mid x))^t\big]$$
$$= -\mathbb{E}_Y\big[(1 - p(Y \mid x))^t \, p(Y \mid x)\big].$$

For each $y$, $(1 - p(y \mid x))^t$ is decreasing in $t$. Multiplying by the positive $p(y \mid x)$ preserves this property, and expectation is linear; therefore the sequence $g_t(x) := \mathbb{E}_Y[(1 - p(Y \mid x))^t p(Y \mid x)]$ is non-increasing, so $\Delta(x,t) = -g_t(x)$ is non-decreasing. $\qquad\square$

## B.2  Proof of Theorem 3.2

Let's start by restating the optimisation problem: For every input $x \in \mathcal{X}$ the fixed query policy $T : \mathcal{X} \to \mathbb{N}$ returns the random multiset $Z(x) = Z\big(T(x), x\big) = \{y_1^x, \ldots, y_{T(x)}^x\}$. A set map $f$ outputs the prediction set $C(x) = f\big(x, Z(x)\big) \subseteq Z(x) \cup \{\text{EE}\}$. The goal is

$$\min_f \quad \mathbb{E}\left[\lambda \, \mathbb{1}\{\text{EE} \in C(X)\} + \sum_{y \neq \text{EE}} \mathbb{1}\{y \in C(X)\}\right] \tag{6}$$
$$\text{s.t.} \quad \Pr\big[Y \in C(X)\big] \geq 1 - \alpha.$$

Let us first outline the strategy for the proof clearly. The optimization problem (4) involves selecting subsets of labels to minimize the frequency of including the abstract label EE and the size of the prediction sets, subject to a coverage constraint. To solve this precisely, we begin by introducing a relaxation to a linear programming problem, argue strong duality and optimality conditions, and then show the relaxation introduces no strictly better fractional solutions, hence the relaxation is actually equivalent to the original problem. Finally, we identify the optimal solution explicitly and demonstrate it has the threshold-based structure stated in the theorem.

**Relaxation to a Linear Program.**  For each $x \in \mathcal{X}$ and realized set $Z(x)$, define a selection variable,
$$g(x, Z(x), y) \in [0, 1], \quad y \in Z(x) \cup \{\text{EE}\}$$

which represents the probability of including label $y$ in the prediction set for covariate $x$ and sampled set $Z(x)$. Replacing $f$ by $g$ and allowing the full interval $[0, 1]$, the optimization problem (4) can then be relaxed to:

$$\min_g \quad \mathbb{E}\left[\lambda \, g(X, Z(X), \text{EE}) + \sum_{y \neq \text{EE}} g\big(X, Z(X), y\big)\right] \tag{7}$$
$$\text{s.t.} \quad \mathbb{E}\big[g\big(X, Z(X), Y\big)\big] \geq 1 - \alpha,$$

This relaxation enlarges the feasible region, i.e., its feasible region contains that of the discrete problem (6) (simply restrict $g$ to $\{0, 1\}$), hence the optimal value of (7) is *no larger* than the optimum of the original integer-valued problem (6).

Both objective and constraint are linear in $g$, so (LP) is a linear programme. In particular, This is a linear programme with one linear constraint, identical in form to the Neyman–Pearson allocation problem. The classical lemma (see, [68] for the case of finite dimensional optimization and Theorem 1, Section 8.3 of [69] for infinite dimensional optimization) states that an optimal solution is obtained by selecting those labels with largest benefit–to–cost ratio until the coverage constraint is met, possibly randomizing on a single tie. As we assumed that there is no mass-point in the underlying distribution, tie-breaking randomization is not necessary, a situation that similarly arises in the original derivation of Neyman–Pearson lemma.

Here the benefit of label $y$ (EE or not) is $p(y \mid x)$. However, the cost is 1 when $y \neq \text{EE}$ and $\lambda$ when $y = \text{EE}$. The benefit–to–cost ratio ordering is therefore equivalent to ordering by the *non-conformity score*

$$S_0(x, y) := \begin{cases} 1 - p(y \mid x), & y \neq \text{EE}, \\ 1 - \frac{p(\text{EE}\mid x)}{\lambda}, & y = \text{EE}. \end{cases}$$

As a result of Neyman–Pearson lemma, there exists a threshold $q_0^* \in \mathbb{R}$ such that,

$$g^\star(x, Z, y) := \mathbf{1}\left\{S_0(x, y) \leq q_0^*\right\}, \tag{8}$$

where $g^*$ is the optimal solution to (7). This automatically results that the relaxed optimization problem (7) is equivalent to the original integer problem (6), as the optimal solution to (7) is of the integer form. That is to say, $f^* := g^*$ is also the optimal solution to (6). We now focus on $g^*$ and show that one can rewrite the same decision rule in the form that is described in Theorem 3.2.

The decision rule, $g^*$, depends solely on the level sets of $S_0$. Here, the key observation is the set of selected labels depends on the level-sets of the function $g^*$, rather than the values it takes. We may therefore apply any strictly decreasing transformation to $S_0$ without changing the selected labels. First, translate the EE row by $+1$ to obtain

$$S_1(x, y) := \begin{cases} 1 - p(y \mid x), & y \neq \text{EE}, \\ 2 - \frac{p(\text{EE}|x)}{\lambda}, & y = \text{EE}. \end{cases}$$

To ensure this transformation does not interfere with the ordering of the original labels, we require that $\lambda$ is sufficiently large. This guarantees that for any $y \neq \text{EE}$, we have $1 - \dfrac{p(\text{EE} \mid x)}{\lambda} > 1 - p(y \mid x)$, so the EE score in $S_0$ is strictly greater than the scores assigned to any concrete label (here we also used the fact that $p(y \mid x) > 0$, which is true as $y$ is one of the "seen" samples, hence the probability of it should be non-zero). Then, shifting the EE score by $+1$ preserves the separation of score ranges: all concrete labels lie in $(0, 1]$ and EE lies in $(1, 2]$.

Next, apply the strictly decreasing map $t \mapsto 2 - \lambda(2 - t)$ on $(1, 2]$; this leaves the concrete labels untouched and sends the EE score to $2 - p(\text{EE} \mid x)$. The resulting score

$$S(x, y) := \begin{cases} 1 - p(y \mid x), & y \neq \text{EE} \\ 2 - p(\text{EE} \mid x), & y = \text{EE} \end{cases}$$

induces exactly the same selection rule and matches (5). That is, the optimal solution to is of the form: $\{y : S(x, y) \leq q^\star\}$ for some $q^* \in \mathbb{R}$. This concludes the Theorem 3.2.

### B.3  Proof of Theorem 4.1

**Proof of Theorem 4.1  (Coverage Validity).**

Define the conformity scores:

$$s_i = \hat{S}(X_i, Y_i), \quad \forall (X_i, Y_i) \in \mathcal{D}_{\text{cal}_2}, \quad \text{and} \quad s_{\text{test}} = \hat{S}(X_{\text{test}}, Y_{\text{test}}).$$

The prediction set is defined as:

$$C(X_{\text{test}}) = \{y \in Z(X_{\text{test}}) \cup \{\text{EE}\} : \hat{S}(X_{\text{test}}, y) \leq q^*\}, \quad \text{where} \quad q^* = \text{Quantile}_{1-\alpha}(s_1, \ldots, s_{N_2}, \infty).$$

We now derive a chain of equalities and inequalities:

$$\Pr[Y_{\text{test}} \in C(X_{\text{test}})] \stackrel{(a)}{=} \Pr[s_{\text{test}} \leq q^*] \stackrel{(a)}{=} \Pr\left[s_{\text{test}} \leq \text{Quantile}_{1-\alpha}(s_1, \ldots, s_{N_2}, \infty)\right]$$

$$\stackrel{(b)}{=} \mathbb{E}\left[\frac{1}{N_2 + 1} \sum_{i=1}^{N_2+1} \mathbb{I}\left[s_i \leq \text{Quantile}_{1-\alpha}(s_1, \ldots, s_{N_2}, s_{\text{test}})\right]\right]$$

$$\stackrel{(c)}{\geq} 1 - \alpha,$$

where,

(a) By definition of the prediction set and $q^*$.

(b) Follows from exchangeability of the scores $\{s_1, \ldots, s_{N_2}, s_{\text{test}}\}$, since $(X_{\text{test}}, Y_{\text{test}})$ is exchangeable with the calibration pairs.

(c) By definition of the $(1 - \alpha)$ quantile, at least a $1 - \alpha$ fraction of the $N_2 + 1$ values are less than or equal to it.

Therefore, we conclude:

$$\Pr[Y_{\text{test}} \in C(X_{\text{test}})] \geq 1 - \alpha,$$

as required. $\qquad\square$

# C Further Experiments and Details

## C.1 Sub-optimal calibration procedure

In our fine-grained, component-wise comparisons, we employ a simple yet valid calibration rule to ensure empirical coverage at the target level $1 - \alpha$. This serves as a sub-optimal but interpretable baseline for evaluating the contributions of each algorithmic principle.

To calibrate, we perform a grid search over a set of candidate thresholds $\{\tau_1, \ldots, \tau_m\} \subset [0, 1]$, uniformly spaced across the interval. For each candidate threshold $\tau_i$, we apply the following two-step procedure on the calibration data $(x_i, y_i)_{i=1}^n$: (i) include the fallback EE cluster in the prediction set if its estimated probability satisfies $\mathbb{P}(\text{EE}) \geq \tau_i$. (ii) sort the remaining clusters by their probabilities in descending order, and sequentially add them to the prediction set until the cumulative probability mass exceeds $1 - \tau_i$. We then compute the empirical coverage at each threshold:

$$cov(\tau_i) = \frac{1}{n} \sum_{i=1}^{n} \{y_i \in C_{\tau_i}(x_i)\}$$

where $C_{\tau_i}(x_i)$ denotes the prediction set constructed with threshold $\tau_i$. We choose $\tau^* = \min\{\tau \in \{\tau_1, \ldots, \tau_m\} : cov(\tau_i) \geq 1 - \alpha\}$.

At test time, we construct prediction sets using the calibrated threshold $\tau^*$ via the same two-step strategy: include EE if its predicted probability satisfies $\mathbb{P}(\text{EE}) \geq \tau^*$, and then add remaining non-EE clusters in order of decreasing probability until the cumulative mass exceeds $1 - \tau^*$.

## C.2 Performance across different budget values

To assess the robustness of each algorithmic component under varying resource limits, we conduct experiments at two additional budget levels for every dataset. These settings are chosen to span regimes where additional queries provide substantial gains (low budget) versus diminishing returns (high budget). Figure 2 shows that in all settings, progressively adding adaptive optimal querying (principle 1) and conformal calibration (principle 2) consistently improves or maintains performance relative to the vanilla baseline. Notably, the largest reductions in EE–fraction occur under tighter budget constraints—when the average number of queries per input is small relative to the model's inherent uncertainty and the difficulty of the dataset. In these regimes, adaptive querying provides the greatest benefit by allocating queries more strategically, thus increasing the likelihood of observing informative labels. In contrast, when the budget is generous enough that most correct answers are already revealed through uniform sampling, the marginal gains from adaptive querying diminish—but are never harmful.

These results collectively reinforce that CPQ delivers targeted gains with the addition of each optimal modular component.

## C.3 Additional comparison with baselines

In this section, we provide further comparison of our algorithm CPQ with CLM and SCOPE-Gen across varying nominal coverage levels for each dataset. Since scope-gen and CLM do not explicitly control the query budget; their number of queries varies depending on the dataset and the desired coverage level. To ensure a fair comparison under shared resource constraints, we first compute the average number of queries used by both CLM and SCOPE-Gen at each coverage level, and configure CPQ to operate under the minimum of these two query budgets. While this setup may disadvantage CPQ in cases where a baseline uses a larger query budget, Table 2 shows that CPQ still consistently achieves tighter empirical coverage and lower EE fractions.

## C.4 Robustness under high hallucination rates

LLMs are inherently prone to hallucinations. While all of our experiments involve black-box LLMs which already exhibit nontrivial hallucination behavior, we additionally evaluate robustness under settings with elevated hallucination rates. To this end, in this section we consider SimpleQA, an adversarial benchmark specifically designed to expose hallucination failures in GPT models.

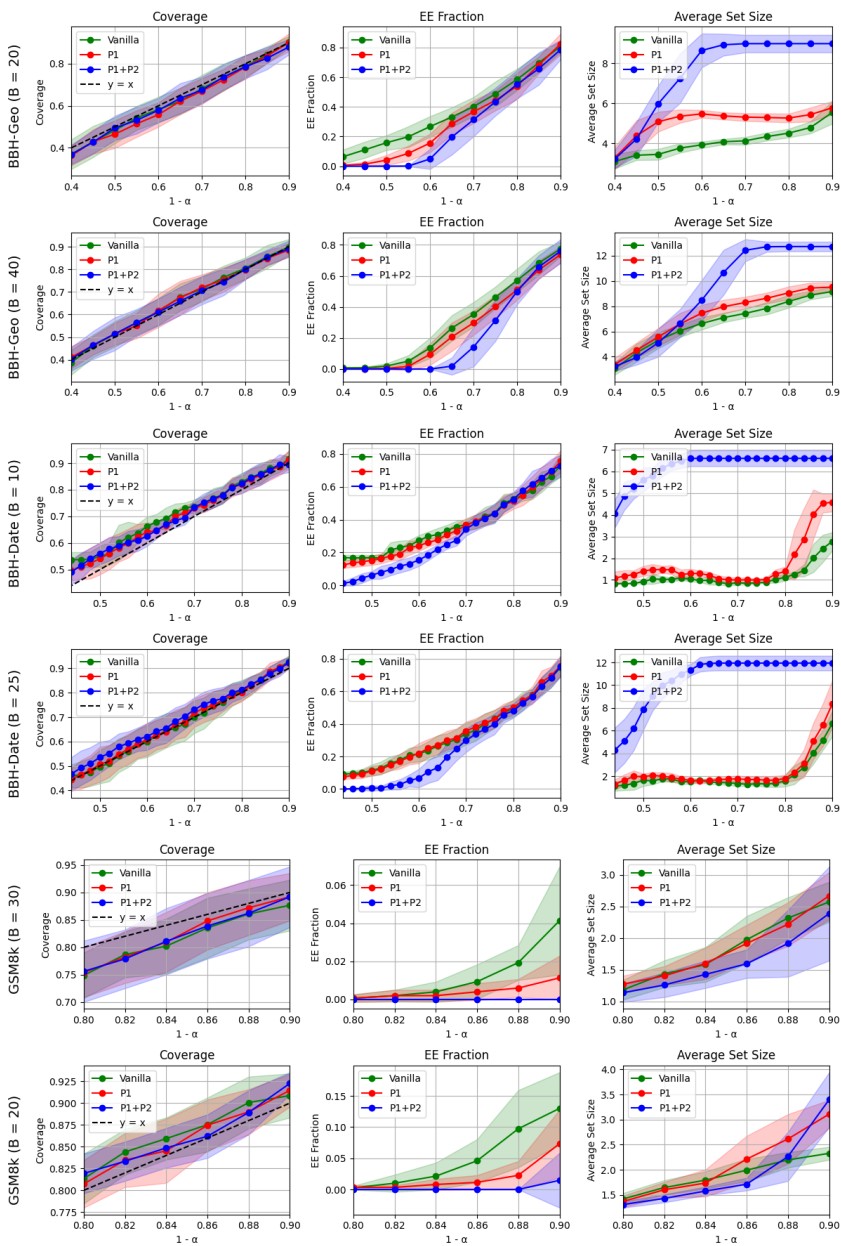

Figure 2: Comparison of the fine-grained variants—vanilla baseline, optimal adaptive querying strategy (Principle 1), and full CPQ (Principles 1 + 2)—under two different budget levels for each dataset. For BBH-Geometric Shapes, the corresponding budget levels are 20 and 40 ; for BBH Date Understanding, 10 and 25; and for GSM8K, 20 and 30. Shaded regions correspond to the standard deviation over ten independent runs.

| Dataset | Algorithm | Nom. Cov. | Emp. Cov. | EE Frac. |
|---|---|---|---|---|
| **GSM8K** | | | | |
| | CLM | 0.97 | $0.93 \pm 0.02$ | $0.74 \pm 0.09$ |
| | Scope-Gen | 0.97 | $0.93 \pm 0.05$ | $0.56 \pm 0.32$ |
| | CPQ | 0.97 | $0.96 \pm 0.02$ | $\mathbf{0.48} \pm 0.15$ |
| | CLM | 0.90 | $0.89 \pm 0.05$ | $0.54 \pm 0.14$ |
| | Scope-Gen | 0.90 | $0.86 \pm 0.06$ | $0.10 \pm 0.12$ |
| | CPQ | 0.90 | $0.89 \pm 0.03$ | $\mathbf{0.00} \pm 0.00$ |
| | CLM | 0.85 | $0.86 \pm 0.05$ | $0.48 \pm 0.04$ |
| | Scope-Gen | 0.85 | $0.85 \pm 0.07$ | $0.01 \pm 0.02$ |
| | CPQ | 0.85 | $0.84 \pm 0.04$ | $\mathbf{0.00} \pm 0.00$ |
| | CLM | 0.80 | $0.83 \pm 0.03$ | $0.48 \pm 0.04$ |
| | Scope-Gen | 0.80 | $0.84 \pm 0.02$ | $0.00 \pm 0.00$ |
| | CPQ | 0.80 | $0.79 \pm 0.03$ | $\mathbf{0.00} \pm 0.00$ |
| **BBH - Geometric Shapes** | | | | |
| | CLM | 0.90 | $0.88 \pm 0.05$ | $0.77 \pm 0.05$ |
| | Scope-Gen | 0.90 | $0.95 \pm 0.03$ | $0.93 \pm 0.05$ |
| | CPQ | 0.90 | $0.90 \pm 0.03$ | $\mathbf{0.76} \pm 0.06$ |
| | CLM | 0.80 | $0.73 \pm 0.08$ | $0.60 \pm 0.09$ |
| | Scope-Gen | 0.80 | $0.85 \pm 0.04$ | $0.76 \pm 0.04$ |
| | CPQ | 0.80 | $0.81 \pm 0.05$ | $\mathbf{0.52} \pm 0.10$ |
| | CLM | 0.70 | $0.65 \pm 0.07$ | $0.49 \pm 0.08$ |
| | Scope-Gen | 0.70 | $0.80 \pm 0.05$ | $0.70 \pm 0.08$ |
| | CPQ | 0.70 | $0.70 \pm 0.06$ | $\mathbf{0.20} \pm 0.12$ |
| | CLM | 0.50 | $0.42 \pm 0.08$ | $0.21 \pm 0.06$ |
| | Scope-Gen | 0.50 | $0.58 \pm 0.10$ | $0.12 \pm 0.19$ |
| | CPQ | 0.50 | $0.50 \pm 0.07$ | $\mathbf{0.00} \pm 0.00$ |
| **BBH - Date Understanding** | | | | |
| | CLM | 0.90 | $0.84 \pm 0.06$ | $0.63 \pm 0.08$ |
| | Scope-Gen | 0.90 | $0.96 \pm 0.05$ | $0.92 \pm 0.10$ |
| | CPQ | 0.90 | $0.90 \pm 0.03$ | $\mathbf{0.72} \pm 0.06$ |
| | CLM | 0.80 | $0.72 \pm 0.10$ | $0.41 \pm 0.12$ |
| | Scope-Gen | 0.80 | $0.88 \pm 0.04$ | $0.71 \pm 0.05$ |
| | CPQ | 0.80 | $0.81 \pm 0.04$ | $\mathbf{0.47} \pm 0.06$ |
| | CLM | 0.60 | $0.52 \pm 0.08$ | $0.12 \pm 0.05$ |
| | Scope-Gen | 0.60 | $0.68 \pm 0.06$ | $0.33 \pm 0.08$ |
| | CPQ | 0.60 | $0.61 \pm 0.06$ | $\mathbf{0.06} \pm 0.04$ |
| | CLM | 0.50 | $0.45 \pm 0.08$ | $0.05 \pm 0.05$ |
| | Scope-Gen | 0.50 | $0.60 \pm 0.08$ | $0.17 \pm 0.05$ |
| | CPQ | 0.50 | $0.51 \pm 0.08$ | $\mathbf{0.00} \pm 0.01$ |

Table 2: Comparison of CPQ with CLM and SCOPE-Gen across nominal coverage levels on GSM8K, BBH–Geometric Shapes, and BBH–Date Understanding. CPQ is constrained to the lowest average query budget used by the baselines at each coverage level. Despite this restriction, CPQ maintains tighter empirical coverage and lower EE fractions.

Table 3 reports our results on GPT-4o generations under this high-hallucination setting. Even in this challenging regime, we observe consistent improvement using our method, following the same trends observed in the main body.

| Algorithm | $1 - \alpha$ | Emp Cov $\pm$ Std | EE $\pm$ Std | Avg Set Size $\pm$ Std |
|---|---|---|---|---|
| Vanilla | 0.60 | $0.62 \pm 0.04$ | $0.16 \pm 0.03$ | $0.85 \pm 0.02$ |
| P1 | 0.60 | $0.59 \pm 0.06$ | $0.11 \pm 0.03$ | $1.20 \pm 0.14$ |
| **P1 + P2 (CPQ)** | **0.60** | **$0.61 \pm 0.05$** | **$0.04 \pm 0.04$** | **$3.47 \pm 0.53$** |
| Vanilla | 0.65 | $0.65 \pm 0.06$ | $0.19 \pm 0.05$ | $0.89 \pm 0.07$ |
| P1 | 0.65 | $0.63 \pm 0.06$ | $0.16 \pm 0.05$ | $1.28 \pm 0.14$ |
| **P1 + P2 (CPQ)** | **0.65** | **$0.65 \pm 0.05$** | **$0.11 \pm 0.06$** | **$2.89 \pm 0.67$** |
| Vanilla | 0.70 | $0.68 \pm 0.07$ | $0.23 \pm 0.06$ | $0.89 \pm 0.03$ |
| P1 | 0.70 | $0.68 \pm 0.05$ | $0.22 \pm 0.04$ | $1.04 \pm 0.04$ |
| **P1 + P2 (CPQ)** | **0.70** | **$0.69 \pm 0.04$** | **$0.18 \pm 0.06$** | **$1.93 \pm 0.58$** |
| Vanilla | 0.80 | $0.82 \pm 0.06$ | $0.48 \pm 0.10$ | $1.15 \pm 0.21$ |
| P1 | 0.80 | $0.79 \pm 0.06$ | $0.41 \pm 0.11$ | $1.77 \pm 0.80$ |
| **P1 + P2 (CPQ)** | **0.80** | **$0.79 \pm 0.05$** | **$0.40 \pm 0.09$** | **$1.70 \pm 0.18$** |

Table 3: Performance under high hallucination rates (SimpleQA, GPT-4o)

## C.5 Clustering algorithm

To group semantically equivalent answers, we apply a relaxed clustering procedure based on pairwise entailment checks using LLaMA-3-8B [35]. Given a question $x$ and two candidate responses $y_1$ and $y_2$, we query LLaMA-3-8B twice: once to determine whether $y_1$ entails $y_2$, and once for the reverse direction. We declare two responses as a match under a relaxed bidirectional entailment criterion: one direction must return `entailment`, and the other must return either `entailment` or `neutral`. This relaxation tolerates mild asymmetries when one answer adds detail without changing the core meaning. Using this matching function, we construct clusters through a simple iterative merging process. Each response is compares against existing clusters, and added to the first cluster containing a match; otherwise it initiates a new cluster. This bucket-merge strategy, while simple, produced highly coherent clusters in practice and was robust across datasets. We emphasize that CPQ is agnostic to the particular clustering routine used. Any method that produces coherent and valid clusters—whether heuristic, learned, or rule-based—can be substituted.

Below we provide the exact system and user prompts used for LLaMA entailment checks, followed by the pseudo code for our relaxed clustering procedure:

```
System:
You are an expert at determining semantic entailment between answers to questions.
Given a question and two answers, determine if Answer 1 entails Answer 2.
Respond with only one word:
entailment, contradiction, or neutral.

User:
Question: <QUESTION>
Answer 1: <RESP1>
Answer 2: <RESP2>

Does Answer 1 semantically entail Answer 2?
```

**Algorithm 2** Relaxed Entailment Clustering

**Input:** question $x$, responses $\{y_i\}_{i=1}^T$

---

MATCH Function (via LLaMA)

1: **function** MATCH$(x, a, b)$
2:     ent1 $\leftarrow$ LLaMAEntail$(x, a, b)$
3:     ent2 $\leftarrow$ LLaMAEntail$(x, b, a)$
4:     **return** (ent1 == entailment and ent2 $\in$ {entailment, neutral})
                           or (ent2 == entailment and ent1 $\in$ {entailment, neutral})
5: **end function**

---

Clustering

- Initialize empty cluster set: $\mathcal{C} \leftarrow \emptyset$
- **for each** response $y_i \in \{y_1, \ldots, y_T\}$:
  - **if** $\exists c \in \mathcal{C}, y \in c$ such that MATCH$(x, y_i, y)$ returns True:    add $y_i$ to cluster $c$
  - **else:** create new cluster $\{y_i\}$ and add it to $\mathcal{C}$

---

**Output:** clusters $\mathcal{C}$

---

### C.5.1 Comparison of alternative clustering methods

Although clustering is not the primary focus of our work, it serves as a necessary pre-processing step when applying our method to natural language outputs. To ensure that our conclusions are not sensitive to the particular clustering strategy used, we conducted an ablation study comparing several representative methods on the *TriviaQA* dataset and evaluated their effect on our algorithm's downstream performance. We considered the following clustering methods:

- **LLaMA-3-8B Entailment (default)**: Pairwise entailment queries between responses using LLaMA-3-8B.

- **W2V:** Averaging 300-dimensional Word2Vec embeddings per response and clustering with KMeans

- **Cosine similarity:** Embedding each response with a MiniLM encoder and linking responses with cosine similarity $\geq 0.85$.

- **NLI-based:** using a fine-tuned RoBERTa-Large model instead for entailment.

We evaluate across two metrics, namely pairwise **overlap** and **Jaccard similarity** (intersection-over-union of clusters across methods), as well as downstream CP metrics (coverage, EE fraction) to evaluate the impact of clustering variation on the behavior of our algorithm. As shown in Table 4, all clustering methods produced highly consistent partitions of the model outputs, with pairwise overlaps exceeding $0.88$ and Jaccard similarities above $0.80$. The downstream CP results in Table 5 confirm that these differences in clustering had only a minimal effect on the final results. Across all methods, the coverage and EE fraction difference remained statistically insignificant. These results suggest that as long as semantically similar outputs are grouped reasonably well, the specific clustering method has little effect on overall performance. That said, clustering remains an active research area, especially for very long-form or domain-specific generations, and users can leverage advances in NLP to fit their particular use cases.

## D  Missing Mass and Missing Mass Derivative

In this section, we will first derive an estimator for the missing mass derivative introduced in Section 4, and then empirically evaluate its performance on two synthetic distributions.

| Method Pair | Overlap | Jaccard |
|---|---|---|
| RoBERTa vs Cosine-MiniLM | 0.91 | 0.91 |
| RoBERTa vs W2V | 0.90 | 0.83 |
| RoBERTa vs LLaMA | 0.88 | 0.81 |
| Cosine-MiniLM vs W2V | 0.95 | 0.82 |
| Cosine-MiniLM vs LLaMA | 0.94 | 0.83 |
| W2V vs LLaMA | 0.93 | 0.87 |

Table 4: Average Pairwise Overlap and Jaccard Similarity Between Clustering Methods

| Clustering Method | $1 - \alpha$ | Coverage | EE Fraction |
|---|---|---|---|
| RoBERTa | 0.8 | $0.81 \pm 0.03$ | $0.42 \pm 0.05$ |
| | 0.6 | $0.61 \pm 0.02$ | $0.09 \pm 0.02$ |
| W2V | 0.8 | $0.81 \pm 0.02$ | $0.44 \pm 0.04$ |
| | 0.6 | $0.62 \pm 0.03$ | $0.10 \pm 0.02$ |
| Cosine-MiniLM | 0.8 | $0.82 \pm 0.02$ | $0.44 \pm 0.05$ |
| | 0.6 | $0.62 \pm 0.03$ | $0.09 \pm 0.02$ |
| **LLaMA (default)** | **0.8** | **$0.80 \pm 0.02$** | **$0.42 \pm 0.02$** |
| | 0.6 | $0.61 \pm 0.02$ | $0.08 \pm 0.01$ |

Table 5: Performance of CPQ with Different Clustering Methods on TriviaQA (LLaMA-3-8B Generations)

## D.1 Derivation

In this section, we study the problem of estimating the missing mass and its rate of change. We abstract away from any specific context (such as input $x$) and define the missing mass problem in a general form. The missing mass is the probability of observing a previously unseen label if we were to draw one additional sample after observing $t$ i.i.d. samples from a discrete distribution. The classical Good–Turing estimator addresses this problem. Here, we derive an estimator for the derivative of the missing mass, which quantifies the rate at which the mass of unseen labels is shrinking as more samples are collected.

We begin by introducing some key quantities and explaining a generative process that mirrors the derivation of the classical Good–Turing estimator, following the notation and exposition from [70]. We then use similar principles to derive an estimator for the rate of change in the missing mass.

Let $\mathcal{Y}$ be the label space, and $W$ denote the sequence of $T$ independent samples $W = \{w_1, \ldots, w_t\}$ where $w_k \in \mathcal{Y}$. Let $\theta_j$ be the probability that a future sample will be $y_j$, where we'd like to account for the probability of $y_j$ occurring even if it has not appeared in the sample $W$. Thus, a simple frequency $\frac{\#(y_j)}{T}$ does not suffice, where $\#(y_j)$ is defined as the number of times label $y_j \in \mathcal{Y}$ appears in $W$. Throughout this derivation, we assume that $\theta_j = \theta_{j'}$ if $\#(y_j) = \#(y_{j'})$, thus two samples appear the same amount of times if they have the same probability of occurring. This assumption is also needed for the classical derivation of the Good-Turing estimator. Though not realistic, this assumption reduces the number of parameters significantly.

Let $N_r = |\{y_j : \#(y_j) = r\}|$ be the number of labels that occur exactly $r$ times in $W$. Let $\theta(r)$ denote the probability of a label occuring given that it appeared $r$ times in $W$. To derive an estimate for $\theta(r)$, consider the following generative process: assume we have access to $\theta_j$. Draw $j$ and hence also $\theta_j$ uniformly at random from the label space $\mathcal{Y}$. Then. flip a coin $t$ times, where $\theta_j$ is the probability of success. Then the number of successes is the number of times $y_j$ appears. if $y_j$ appears $r$ times, put $\theta_j$ in $\theta(r)$. At the end $\theta(r)$ will approximately be the average of the $\theta_j$ for which $\#(y_j) = r$.

Precisely

$$\hat{\theta}(r) = \mathbb{E}\big[\theta_j | \#(y_j) = r\big] = \sum_j \theta_j \mathbb{P}\big[\theta_j | \#(y_j) = r\big]$$

Now, condition on $\theta_j$ by applying Bayes rules , and given the uniform prior on $\mathbb{P}(\theta_j) = \frac{1}{m}$, we obtain the following for the probability of a $y_j$ appearing given that it has appeared $r$ times is

$$\frac{\sum_j \theta_j \, \mathbb{P}\big[\#(y_j) = r \mid \theta_j\big]}{\sum_{j'} \theta_{j'} \, \mathbb{P}\big[\#(y_{j'}) = r \mid \theta_{j'}\big]}$$

We can rewrite both the numerator and the denominator in terms of the pdf of the binomial distribution:

$$\frac{\sum_j \theta_j \binom{t}{r} \theta_j^r (1 - \theta_j)^{t-r}}{\sum_{j'} \theta_{j'} \binom{t}{r} \theta_{j'}^r (1 - \theta_{j'})^{t-r}}$$

We can rewrite the denominator in terms of $\mathbb{E}_{\text{in } t}[N_r]$, the expected value of $N_r$ given that we flipped $t$ coins at each step of our experiments, yielding the following equation:

$$\frac{1}{\mathbb{E}_{\text{in } t}[N_r]} \sum_j \theta_j \binom{t}{r} \theta_j^r (1 - \theta_j)^{t-r}$$

This quantity is estimating the *probability of a label* conditioned on it appearing exactly $r$ times in the sample—that is, the expected value of $\theta_j$ given $\#(y_j) = r$. However, what we actually want is the *total probability mass* of all such labels. To obtain that, we need to multiply the average by the *number of labels* that appeared $r$ times. Notably, the denominator of the expression we derived is $\mathbb{E}[N_r]$, the expected number of such labels. So in fact, the numerator alone gives an estimation of the total probability mass.

Furthermore, we'd like to derive and estimate of the change in missing mass, we set $r = 0$, thus we are interested in the following quantity:

$$\sum_j \theta_j (1 - \theta_j)^{t+1} - \sum_j \theta_j (1 - \theta_j)^t = \sum_j -\theta_j^2 (1 - \theta_j)^t$$

$$= \frac{-2}{(t+2)(t+1)} \sum_j \binom{t+2}{2} \theta_j^2 (1 - \theta_j)^t$$

$$\overset{(a)}{=} \frac{-2}{(t+2)(t+1)} \mathbb{E}_{\text{in } t+2}\big[N_2\big]$$

$$\overset{(b)}{\approx} \frac{-2N_2}{t^2}$$

where (a) follows from the fact that $\mathbb{E}_{\text{in } t+2}\big[N_2\big] = \sum_j \binom{t+2}{2} \theta_j^2 (1 - \theta_j)^t$ which is due to a simple counting argument. (b) is due to an approximation for sufficiently large $t$, and plugging $N_2$ as $\mathbb{E}_{\text{in } t+2}\big[N_2\big]$.

Hence, this yields our proposed estimator introduced in Section 4 for the missing mass rate of decay

$$\boxed{\hat{\Delta}(t) = \frac{-2N_2}{t^2}}$$

.

## D.2 Empirical evaluation

We conduct experiments on two synthetic distributions over a support of size 100: (i) a uniform distribution, $\pi_i = 1/100$ for all $i$, and (ii) a geometric distribution, $\pi_i = p(1 - p)^{i-1}$ with $p = 0.05$. Figure 3 presents two panels for each distribution. In the left panels, we compare the true missing mass $\theta(t)$ (red dashed) against the Good–Turing estimate $\hat{\theta}(t)$ (blue solid). In the right panels, we compare the true derivative (red dashed) against our proposed derivative estimator $\hat{\Delta}(t) = \frac{-2N_2}{t^2}$ (blue) and the naive finite-difference of the Good-Turing estimator baseline $\hat{\Delta}(t) = \hat{\theta}(t+1) - \hat{\theta}(t)$ (Green). Across both distributions, the Good–Turing estimator closely tracks the ground truth and its variance decays as more observations are collected. Similarly, our estimator closely captures the decay rate of the missing-mass derivative with substantially lower variance and fluctuations than the naive difference-based baseline.

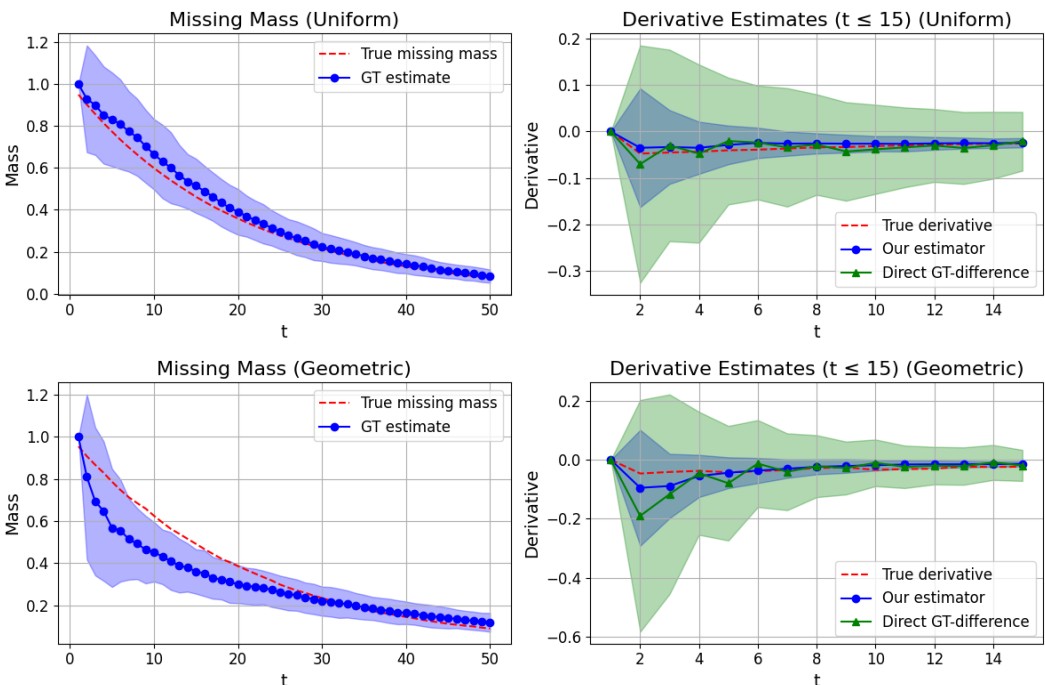

Figure 3: Empirical comparison of missing mass and its derivative estimators on two synthetic distributions: uniform (top panels) and geometric with $p = 0.05$ (bottom panels). *Left panels:* true missing mass (red dashed line) versus the Good–Turing estimator (blue solid line). *Right panels:* true derivative (red dashed line) compared to our proposed derivative estimator (blue) and the naive finite-difference baseline (green). The standard deviation after averaging across 100 independent trials is represented by the shaded region in each corresponding color.

