# OpenReview forum: "Conformal Prediction Beyond the Seen: A Missing Mass Perspective for Uncertainty Quantification in Generative Models"
_NeurIPS.cc/2025/Conference — NeurIPS 2025 poster_

### Official Review · Reviewer_tHYJ · 2025-06-23

**Clarity:** 4
**Significance:** 3
**Originality:** 3
**Rating:** 5
**Confidence:** 2

**Summary:**

The paper introduces CPQ, a framework for UQ in the query-only setting where prediction sets are constructed from queries to a generative model. CPQ is based on two components: a query policy, and a set mapping function. Notably, the query policy is based on the rate of decrease of the missing mass. The authors show on three tasks that CPQ yields informative predictions sets.

**Questions:**

- is the clustering robust enough in practice? for example, what happens if you change the LLM used to define the semantic equivalences, and what's the impact on the CP metrics
- what happens in the low query regime?

**Ethical Concerns:**

["NO or VERY MINOR ethics concerns only"]

**Final Justification:**

I keep my recommendation for acceptance. The authors have provided detailed rebuttals that have addressed my questions.

**Limitations:**

The authors discuss the very low query regime as a possible limitation.

**Quality:**

3

**Strengths And Weaknesses:**

Strengths:
- the paper is very well-written
- it tackles an important problem (UQ when the label space is unstructured, as is the case with text generation) and realistic setting (query only generative setting).
- CPQ has finite sample coverage guarantees
- the experimental results seem to suggest that the prediction sets are informative (based on the set's size and the fraction of EE labels)

Weaknesses:
- $\hat{\Delta}(x,t)$ is based on $N_2$, the number of samples that appear twice. When we don't have a lot of sampling budget, $N_2$ is likely to be $0$, meaning that the stopping condition is achieved. Hence additional results in this low-query setting might be interesting, for example via a sensitivity analysis on the budget. In the low query setting, it would be interesting to have an ablation where the authors use a fixed querying strategy, and their proposed calibration rule (hence a Principle 2 baseline)
- as the authors admit it, the derivation of the estimator for $\hat{\Delta}(x,t)$ builds on quite a strong assumption that all samples appearing the same number of times have the same probability of occurring.
- it would be interesting to check what is happening when $\pi$ greatly differs from the ground truth $p$ (e.g. an LLM prone to hallucinations at some input points).

---

> ### Author Rebuttal · Authors · 2025-07-29
>
> Thank you for the clear and encouraging feedback. We're glad you found the paper well-written and recognized both the theoretical and practical strengths of our approach.
>
> We appreciate your thoughtful suggestions and we have addressed each below:
>
> ## **P2 baseline study: Fixed sampling + Optimal calibration rule** | **Low query regime**
> Thank you for this insightful and interesting suggestion. To evaluate the effectiveness of our **Principle 2** in isolation we conducted an ablation where we used a fixed querying strategy and applied only our proposed optimal calibration rule.
> These results confirm that **our calibration rule is effective even when adaptive querying is infeasible (e.g in very low query regimes),** and ensures our full algorithm (CPQ) remains beneficial under strict query budgets. A detailed report will be added to the revised manuscript.
>
> We ran this study under two settings:
> - Moderate query budget (B = 30)
> - Low query budget (B =5)
>
> Our **Findings**:
> - In both regimes, **P2 alone improves over the vanilla baseline**, both in terms of reducing the prediction set size, and reducing the fraction of EE while maintaining coverage.
> - In the **low query regime**, as you also mentioned, the missing mass derivative becomes unstable, so adaptive querying alone (P1), is less effective. As a result, **P2 dominates** and **P1+P2 (CPQ) performs similarity to P2 only**.
> - In the moderate-query regime, where the derivative estimation is more reliable, combining P1 with P2  (CPQ) yields further gains over either principle in isolation--but **P2-only still provides meaningful improvements.**
>
>  ### **Budget = 30 queries**
>
> | Method   | 1 − α = 0.7 Coverage ± Std | EE ± Std | Size ± Std |   | 1 − α = 0.6 Coverage ± Std | EE ± Std | Size ± Std |
> |----------|----------------------------|----------|-------------|---|----------------------------|----------|-------------|
> | Vanilla  | 0.70 ± 0.06                | 0.40 ± 0.07 | 5.92 ± 0.28 |   | 0.60 ± 0.07                | 0.20 ± 0.08 | 5.47 ± 0.31 |
> | P1       | 0.70 ± 0.06                | 0.29 ± 0.08 | 7.54 ± 0.35 |   | 0.60 ± 0.07                | 0.07 ± 0.06 | 7.23 ± 0.69 |
> | **P2**   | **0.71 ± 0.06**            | **0.31 ± 0.12** | **3.10 ± 1.10** |   | **0.60 ± 0.07**            | **0.03 ± 0.05** | **8.58 ± 1.39** |
> | P1+P2    | 0.71 ± 0.06                | 0.21 ± 0.12 | 7.38 ± 1.78 |   | 0.61 ± 0.07                | 0.01 ± 0.02 | 8.58 ± 1.39 |
>
> ### **Budget = 5 queries – (Low Query Regime)**
>
> | Method   | 1 - α = 0.7 Coverage ± Std | EE ± Std | Size ± Std |   | 1 - α = 0.4 Coverage ± Std | EE ± Std | Size ± Std |   | 1 - α = 0.3 Coverage ± Std | EE ± Std | Size ± Std |
> |----------|------------------------|----------|-------------|---|------------------------|----------|-------------|---|------------------------|----------|-------------|
> | Vanilla  | 0.74 ± 0.06            | 0.59 ± 0.07 | 1.03 ± 0.09 |   | 0.42 ± 0.14            | 0.29 ± 0.15 | 1.05 ± 0.30 |   | 0.38 ± 0.04            | 0.21 ± 0.03 | 0.84 ± 0.15 |
> | P1       | 0.71 ± 0.07            | 0.56 ± 0.09 | 1.43 ± 0.09 |   | 0.40 ± 0.07            | 0.18 ± 0.05 | 1.58 ± 0.11 |   | 0.31 ± 0.06            | 0.09 ± 0.03 | 1.43 ± 0.27 |
> | **P2**   | **0.70 ± 0.06**        | **0.53 ± 0.06** | **0.96 ± 0.18** |   | **0.40 ± 0.09**        | **0.13 ± 0.07** | **2.40 ± 0.17** |   | **0.31 ± 0.08**        | **0.04 ± 0.04** | **2.28 ± 0.27** |
> | P1+P2    | 0.71 ± 0.07            | 0.53 ± 0.09 | 0.93 ± 0.06 |   | 0.40 ± 0.06            | 0.09 ± 0.06 | 3.18 ± 0.38 |   | 0.30 ± 0.06            | 0.004 ± 0.01 | 2.82 ± 0.55 |
>
> ## **On the robustness of clustering**
> To evaluate robustness to clustering, we compared four methods on TriviaQA and assessed their impact on our algorithm’s downstream performance. **We found that as long as semantically similar outputs are reasonably grouped, the specific clustering method had minimal effect.**
>
> Clustering method evaluated:
> - **LLaMA-3 8B Entailment (default):** Pairwise entailment queries between responses using LLaMA-3-8B.
> - **Word2Vec + KMeans:** Averaging 300-dimensional Word2Vec embeddings per response and clustering with KMeans (as suggested by the reviewer).
> - **Cosine similarity:**  Embedding each response with a MiniLM encoder and linking responses with cosine similarity ≥ 0.85.
> - **NLI-based (RoBERTa-Large MNLI):**  using a fine-tuned RoBERTa-Large model instead for entailment.
>
> We report **Pairwise overlap** ( fraction of response pairs clustered together in both methods) and **Jaccard Similarity** ( intersection-over-union of clusters across methods), as well as downstream metrics (empirical coverage, EE fraction, and average set size).
>
> Our **key findings** are:
> - All clustering methods yielded comparable CP metrics; no dramatic performance shifts were observed.
> - On longer or more open-ended generations, LLaMA-3-8B gave the most reliable clusters by manual inspection, and achieved slightly better downstream metrics.
>
> We also evaluated clustering overlap between LLaMA-3-8B and GPT-4o on a small TriviaQA (50 exampels) subset ( reported below). The clusters showed high Jaccard and pairwise similarity. Due to time constraints, GPT-4o was not run on the full dataset, but based on similarity, we expect performance would remain consistent.
>
> **Table 1: Average Pairwise Overlap and Jaccard Similarity Between Clustering Methods**
>
> | Method Pair                  | Overlap | Jaccard |
> |:-----------------------------|------------:|------------:|
> | RoBERTa vs Cosine-MiniLM |     0.91  |     0.91  |
> | RoBERTa vs W2V    |     0.90  |     0.83  |
> | RoBERTa vs LLaMA      |     0.88  |     0.81  |
> | Cosine-MiniLM vs W2V  |     0.95  |     0.82  |
> | Cosine-MiniLM vs LLaMA   |     0.94  |     0.83  |
> | W2V vs LLaMA       |     0.93  |     0.87  |
> | GPT4o vs LLaMA  | 0.91|   0.89 |
>
> **Table 2: Performance of Different Clustering Methods on TriviaQA (LLaMA-3-8B Generations)**
> | Clustering Method  | 1 - α   | Coverage (mean ± std) | EE Fraction (mean ± std) |
> |:-------------------|:----|:----------------------|:--------------------------|
> | RoBERTa        | 0.8 | 0.81 ± 0.03       | 0.42 ± 0.05           |
> |                    | 0.6 | 0.61 ± 0.02       | 0.09 ± 0.02           |
> | W2V         | 0.8 | 0.81 ± 0.02       | 0.44 ± 0.04           |
> |                    | 0.6 | 0.62 ± 0.03       | 0.10 ± 0.02           |
> | Cosine-MiniLM      | 0.8 | 0.82 ± 0.02      | 0.44 ± 0.05           |
> |                    | 0.6 | 0.62 ± 0.03      | 0.09 ± 0.02           |
> | LLaMA (default) | 0.8 | 0.80 ± 0.02       | **0.42** ± 0.02           |
> |                    | 0.6 | 0.61 ± 0.02      | **0.08** ± 0.01         |
>
> **Take-away**: as long as semantically equivalent outputs are correctly grouped, our algorithm is agnostic to clustering. When using a sufficiently strong model like LLaMA-3-8B, clustering quality is typically not a bottleneck. That said, clustering remains an active research area, especially for very long‐form or domain-specific generations, and users can leverage advances in NLP (from improved embeddings to specialized entailment models) to fit their particular use cases. A detailed report of our findings will be added to the revised version of the paper.
>
> ## **Robustness under hallucination prone setting - when oracle $\pi$ is very different than the ground truth $p$**
> We appreciate the reviewer’s question regarding cases where model outputs may significantly differ from the ground truth (e.g., when the LLM hallucinates). In fact, all of our experiments are conducted with real black-box LLMs (e.g., LLaMA-3, GPT-3.5, GPT-4o), which are inherently prone to hallucinations.
>
> To more directly address this question and evaluate robustness under high hallucination rates, we included SimpleQA, a benchmark adversarially constructed to expose hallucination failures in GPT models. **Even on this challenging dataset, we observed consistent improvements using our method, following the same trends observed in the main paper.**
>
> #### **SIMPLE QA – GPT4o**
>
> | Algorithm           | 1 - α    | Emp Cov ± Std    | EE ± Std         | Avg Set Size ± Std |
> |:--------------------|:----:|------------------:|------------------:|--------------------:|
> | Vanilla             | 0.60 | 0.62 ± 0.04       | 0.16 ± 0.03       | 0.85 ± 0.02         |
> | P1                  | 0.60 | 0.59 ± 0.06       | 0.11 ± 0.03       | 1.20 ± 0.14         |
> | P1 + P2 (CPQ)       | 0.60 | 0.61 ± 0.05       | **0.04 ± 0.04**   | 3.47 ± 0.53         |
> | ---                 |      |                   |                   |                     |
> | Vanilla             | 0.65 | 0.65 ± 0.06       | 0.19 ± 0.05       | 0.89 ± 0.07         |
> | P1                  | 0.65 | 0.63 ± 0.06       | 0.16 ± 0.05       | 1.28 ± 0.14         |
> | P1 + P2 (CPQ)       | 0.35 | 0.65 ± 0.05       | **0.11 ± 0.06**   | 2.89 ± 0.67         |
> | ---                 |      |                   |                   |                     |
> | Vanilla             | 0.70 | 0.68 ± 0.07       | 0.23 ± 0.06       | 0.89 ± 0.03         |
> | P1                  | 0.70 | 0.68 ± 0.05       | 0.22 ± 0.04       | 1.04 ± 0.04         |
> | P1 + P2 (CPQ)       | 0.70 | 0.69 ± 0.04       | **0.18 ± 0.06**   | 1.93 ± 0.58         |
> | ---                 |      |                   |                   |                     |
> | Vanilla             | 0.80 | 0.82 ± 0.06       | 0.48 ± 0.10       | 1.15 ± 0.21         |
> | P1                  | 0.80 | 0.79 ± 0.06       | 0.41 ± 0.11       | 1.77 ± 0.80         |
> | P1 + P2 (CPQ)       | 0.80 | 0.79 ± 0.05       | **0.40 ± 0.09**   | 0.70 ± 0.18         |
>
> ## **On the assumption for deriving missing mass derivative estimator**
> The assumption we make is exactly the one originally introduced and required by Good and Turing for their estimator of missing mass. We adopt the same assumption for estimating the missing mass derivative, as it is well-documented that the Good-Turing estimator performs reliably in practice despite this simplification.

---

> > ### Comment · Reviewer_tHYJ · 2025-08-04
> >
> > Thank you for your detailed rebuttals, I keep my recommendation for acceptance.

---

### Official Review · Reviewer_mpV9 · 2025-06-29

**Clarity:** 4
**Significance:** 4
**Originality:** 4
**Rating:** 5
**Confidence:** 3

**Summary:**

The paper addresses uncertainty quantification of generative models of discrete sequences (where it is not feasible to enumerate the full label set) using conformal inference by introducing Conformal Prediction with Query Oracle (CPQ) which considers the trade-offs between coverage, query budget, and informativeness of the resulting prediction set. The paper introduces an algorithm for querying from the model and producing the prediction set from the queries, unifying both strategies via the missing mass problem.

**Questions:**

If the authors could comment on the above points under Weaknesses, that would be greatly appreciated, but I also understand that addressing some of these comments sufficiently would likely be another paper in itself.

On a separate note, to make the paper more self-contained, could the authors briefly describe the baseline methods as well?

**Ethical Concerns:**

["NO or VERY MINOR ethics concerns only"]

**Final Justification:**

I appreciate the thorough response from the authors to both my review and the other reviews, and my positive review of the work still stands.

**Limitations:**

Yes

**Quality:**

4

**Strengths And Weaknesses:**

Strengths:
- The paper addresses the important question of uncertainty quantification of black-box generative models.
- The paper provides an elegant conceptualization of how to construct the prediction sets and to end, starting from the strategy for querying.
- The paper is clearly written, and the contributions are, to my knowledge, novel.
- Experiments clearly show the benefits of the proposed algorithm.
- The supplement does a good job explaining relevant concepts in the missing mass problem such that this paper can be self contained.

Weaknesses:
- Practically, the algorithm is quite expensive, requiring multiple generations from the model for each prompt. While I understand why this is necessary given the assumptions of the task, i.e., access only to a black box model, this limitation may prevent the use of the method in practice.
- The canonical limitation that coverage is only marginal rather than conditional still applies in the setting, but this issue is a bigger one for conformal inference in general.
- Small nitpick: the objective and the algorithm have the same name, which could get confusing for discussing this work.

---

> ### Author Rebuttal · Authors · 2025-07-28
>
> We sincerely thank the reviewer for their thoughtful and positive feedback. We're especially glad that you found the conceptual framework elegant, the contributions novel, and the paper clearly written. We also appreciate your recognition of the experimental validation and the clarity of the supplemental material in making the paper self-contained.
>
> Below we will address the comments and questions made by the reviewer:
>
> ### **Practical Cost of Querying**
> Thank you for raising this thoughtful point. We agree that repeated querying can be costly in practice, especially for large models. However, there are three arguments here:
>
> 1) **Querying is the only way to uncover the output space in black-box settings:** As you noted, when working with black-box generative models, querying is our only means of exploring the output space. Even with white-box access, for example to an LLM, it remains unclear how one could leverage internal representations to construct prediction sets. It’s important to emphasize that a prediction set must contain multiple fully plausible outputs, and in generative modeling, producing even a single output requires a full round of inference. That being said, computational cost is a central motivation behind our work. Our algorithmic framework is specifically designed to make the most out of a fixed query budget.
>
> 2) **Moderate querying is already common in practice (e.g., best-of-\(N\) sampling):**  Techniques like best-of-\(N\) sampling, where a model is queried multiple times and the best response is selected, are widely used in practical applications. Our framework aligns with this paradigm but provides a principled way to determine how many queries to issue, and when to stop, to make useful prediction sets. The inference time for 10-20 queries lie somewhere between a few seconds (for smaller models) to a few minutes (for larger models) for a prompt in test time. In high-stakes applications such as medical decision-making, the decision-making process often takes more than a few minutes. In such cases, a modest increase in inference time may be well worth the additional reliability it provides.
>
> 3) **Fallback in low-query regimes:**
> Importantly, **when the query budget is very low (e.g. $\leq 5$), one can still use the Principle 2 part of our algorithm alone!** Note that Principle 1 requires access to the value of $N_2$ (the number of double-tones), whereas Principle 2 only depends on $N_1$ (the number of singletons), as it estimates the missing mass using the Good–Turing estimator. Notably, it is well-documented in practice that the Good–Turing estimator based on $N_1$ performs reliably even with very few samples. To illustrate this, we conducted an experiment where the query budget is fixed at 5--querying each test point exactly five times (i.e., using a non-adaptive policy)--and applied only the calibration step guided by Principle 2. The results of this experiment are summarized in the table below ( more details can be found in our response to reviewer tHY). **In short, we still observe meaningful performance gains from applying our calibration module (Principle 2) when querying budget is extremely limited ( eg. <= 5)**. This highlights the practicality of our method even in such restrictive setting.
>
> **Budget = 5 queries – (Low Query Regime)**
>
> | Method   | 1 - α = 0.7 Coverage ± Std | EE ± Std | Size ± Std |   | 1 - α = 0.4 Coverage ± Std | EE ± Std | Size ± Std |   | 1 - α = 0.3 Coverage ± Std | EE ± Std | Size ± Std |
> |----------|------------------------|----------|-------------|---|------------------------|----------|-------------|---|------------------------|----------|-------------|
> | Vanilla  | 0.74 ± 0.06            | 0.59 ± 0.07 | 1.03 ± 0.09 |   | 0.42 ± 0.14            | 0.29 ± 0.15 | 1.05 ± 0.30 |   | 0.38 ± 0.04            | 0.21 ± 0.03 | 0.84 ± 0.15 |
> | P1       | 0.71 ± 0.07            | 0.56 ± 0.09 | 1.43 ± 0.09 |   | 0.40 ± 0.07            | 0.18 ± 0.05 | 1.58 ± 0.11 |   | 0.31 ± 0.06            | 0.09 ± 0.03 | 1.43 ± 0.27 |
> | **P2**   | **0.70 ± 0.06**        | **0.53 ± 0.06** | **0.96 ± 0.18** |   | **0.40 ± 0.09**        | **0.13 ± 0.07** | **2.40 ± 0.17** |   | **0.31 ± 0.08**        | **0.04 ± 0.04** | **2.28 ± 0.27** |
> | P1+P2    | 0.71 ± 0.07            | 0.53 ± 0.09 | 0.93 ± 0.06 |   | 0.40 ± 0.06            | 0.09 ± 0.06 | 3.18 ± 0.38 |   | 0.30 ± 0.06            | 0.004 ± 0.01 | 2.82 ± 0.55 |
>
>
>
> ### **Conditional validity**
> Thank you for bringing this up and also the kind wording. The problem of conditional coverage, as you noted, is an active area of research within the conformal prediction (CP) community and extends beyond CP applications involving LLMs. In this paper, our primary focus was advancing CP specifically for challenges arising in generative modeling, without explicitly addressing conditional coverage.
>
> That said, conditional coverage is an important and promising direction for future work, which we plan to discuss explicitly in the revised manuscript. As a starting point, one could explore integrating our framework with recent techniques proposed by [1], which provides relaxed conditional coverage by replacing the scalar threshold (used in our current method) with a multi-dimensional calibration step. For instance, using their method, one might be able to calibrate a linear model head via quantile regression on representations derived from a pretrained model, thus achieving some levels of conditional validity.
>
> However, carefully exploring the precise implications, applicability, and practical implementation of integrating such methods with the missing mass ideas developed in our paper remains a valuable and open research direction.
>
> [1]: Conformal prediction with conditional guarantees, Gibbs et. al.
>
> ### **Q – Brief Description of Baselines**
> Thank you for the helpful suggestion. Due to space constraints, we originally omitted detailed descriptions of the baselines. However, we agree that including a brief overview would make the paper more self-contained and easier to follow. In the revised version, we will incorporate concise summaries of both **CLM** and **SCOPE-Gen** to clarify how they relate to our setting and evaluation. To describe each method briefly here:
>
> - CLM constructs a prediction set by sequentially sampling outputs from a language model. It stops once a calibrated conformal threshold indicates the set likely contains a correct answer. This threshold is computed during a separate calibration phase to guarantee coverage. However, if the model doesn’t generate the correct answer often enough (e.g., at high coverage levels), the method can’t find a valid stopping threshold and fails to return a set.
>
> - SCOPE-Gen improves on CLM by reducing the number of expensive “admissibility checks” needed during calibration—for instance, when a human expert needs to verify whether an output is valid (as in medical domains). Instead of sampling one-by-one like CLM, it starts with a large batch of outputs and prunes them using a sequence of filters. These filters are designed so that conformal guarantees still apply, but the number of checks needed is much lower.
>
> ### **On the algorithm  and optimization name:**
> Thank you for pointing this out. We really appreciate the thoughtful suggestion and agree that having the objective and algorithm share the same name could lead to confusion. In the revised version, we will rename the objective to better distinguish it from our finite sample algorithm (CPQ), while keeping the terminology aligned with the broader goals of the paper.

---

### Official Review · Reviewer_YqEn · 2025-07-01

**Clarity:** 3
**Significance:** 2
**Originality:** 3
**Rating:** 4
**Confidence:** 3

**Summary:**

This paper introduces CPQ, a framework for UC in generative models to address the challenge
of constructing prediction sets when the true output may not be among sampled generations.
The key innovation is connecting optimal query policies and prediction set construction to the
classical missing mass problem in statistics. The authors introduce an abstract label EE to
represent unseen outputs and derive two algorithmic principles for optimal query and optimal
set construction for finite samples. The finite-sample algorithm maintains distribution-free
coverage guarantees while rely less on uninformative prediction sets compared to existing
methods.

**Questions:**

1. Can you compare to unsupervised clustering method with naïve embedding method
like word2vec? Would it feasible at all or more stable compared to relying on LLM
embeddings?

2. Is the gap between the proposed approximate optimization method significant enough
comparing to join optimization, especially in practice experiments?

3. Is this method robust to shifted domains?

4. Is this method practical/useful on larger models of size 70B+?

5. Can this theory lead to an empirical p-value in terms of confidence, given EE fraction or
with extra assumptions?

**Ethical Concerns:**

["NO or VERY MINOR ethics concerns only"]

**Final Justification:**

The authors have addressed many of my concerns. My assessment of the paper is that it makes a positive contribution, however, I remain unsure whether this rises to the level of acceptance at NeurIPS.

**Limitations:**

Yes

**Paper Formatting Concerns:**

None Noted

**Quality:**

3

**Strengths And Weaknesses:**

Strengths:

The problem formulation addresses a real limitation in existing conformal prediction methods
for generative models. They introduce "EE" label to provide a reasonable way to handle
scenarios where correct answers lie outside sampled sets, which also makes the original
problem somewhat more straightforward.

Theorem 4.1 t to connect the problem to missing mass estimation reads sound and provides
coverage guarantees. Two algorithmic principles seem derived correctly. Though, as
acknowledged by the authors, the decoupled optimization approach as an approximation to
the full joint optimization might be a limiting.

They’ve done fair comparison experiments showing improvement over baselines. They also did
appropriate ablation studies.

Paper is clearly written, providing a useful theoretical lens on perspective of missing mass.


Weakness:

Experimental scope seems limited focusing on just three benchmark datasets. Tasks are
simple and models are small. With the potential, more complex tasks should be tested. Also,
improvement seems minor considering the cost and imperfect oracles in real world, it raises a
question on whether the work is practically impactful.

Authors mentioned clustering is critical which introduced extra indeterminacy, as well
estimation of missing mass can be challenging at low sample situation.

Performance of semantic clustering from LLMs can vary dramatically. It needs a bit more
discussion on the overhead of this part.

Baselines were not originally designed for the same problem setting which makes the
comparison less fair.

---

> ### Author Rebuttal · Authors · 2025-07-28
>
> We thank the reviewer for their detailed and constructive feedback. We're glad they found the problem formulation meaningful, our theoretical perspective sound and useful, and the paper clearly written. Below, we respond to each of your comments and suggestions.
>
> ## **Experimental Scope and Practical Impact + Q4) Practicality for large models 70B+**
> We appreciate your feedback and agree that expanding the experimental scope can enrich our work. To clarify practical significance, we first note that the key metric in conformal prediction (CP) is efficiency, commonly quantified by prediction set size. In our context, two interrelated aspects determine practical utility: the fraction of sets including the EE label ("EE fraction") and the average size of prediction sets.
>
> Including EE signals full uncertainty over the label space, effectively making the set size infinite and the output uninformative. Thus, practical impact is best assessed by first minimizing the EE fraction. When EE fractions are similar (common at high coverage), the focus shifts to minimizing the average set size. Our framework is designed to optimize this trade-off:
>
> - At moderate coverage, it significantly reduces EE inclusion, improving informativeness.
>
> - At high coverage, where EE is unavoidable, it reduces the average set size, improving practical utility.
>
> Moreover, CP efficiency is known to depend on model accuracy. To reflect this, we intentionally designed our experiments to span a wide range of difficulty relative to the models used, covering best-of-N (5) accuracies from below 40% to above 90%, enabling evaluation across both strong and weak models.
>
> That being said, we fully acknowledge the value of broader evaluations to larger-scale models and broader datasets. To further strengthen our experimental results, we have now expanded our evaluation to include:
>
> - **Larger models**: GPT-3.5-turbo, and GPT-4o, supplementing the smaller models used in the original submission.
>
> - **Additional datasets**: Two additional QA benchmarks --TriviaQA, a factual QA dataset, and SimpleQA, an adversarial benchmark notably challenging even for GPT-4o (achieving less than ~30% exact match accuracy in our experiments).
>
> These new results are summarized in the table provided in our response to Reviewer ZMjq (omitted here due to space). A detailed report with plots will be added to the revised version. In short, **across all settings, we observed consistent trends with the results presented in the main paper.**
>
>
> ## **Comparison of Clustering Methods**
> While clustering is not the contribution of our work, it is a necessary pre-processing step in applying our method to natural language outputs. To evaluate robustness to clustering, we compared four clustering methods on TriviaQA ( Table 1 ) and assessed the impact of each method on our algorithm’s downstream performance ( Table 2). **We found that as long as semantically similar outputs are reasonably grouped, the specific clustering method had minimal effect.**
>
> Clustering methods evaluated:
> - **LLaMA-3 8B Entailment (default):**
> - **W2V:** Word2Vec avg + KMeans ( As per reviewer's suggestion)
> - **Cosine similarity:**  MiniLM-embeddings with cosine similarity ≥ 0.85.
> - **NLI-based:**  using a fine-tuned RoBERTa-Large model for entailment.
>
> We report **Pairwise overlap** and **Jaccard Similarity** ( intersection-over-union of clusters across methods), as well as downstream CP metrics ( coverage, EE frac) in the table below.
>
> **Table 1: Average Pairwise Overlap and Jaccard Similarity Between Clustering Methods**
>
> | Method Pair                  | Overlap | Jaccard |
> |:-----------------------------|------------:|------------:|
> | RoBERTa vs Cosine-MiniLM |     0.91  |     0.91  |
> | RoBERTa vs W2V    |     0.90  |     0.83  |
> | RoBERTa vs LLaMA      |     0.88  |     0.81  |
> | Cosine-MiniLM vs W2V  |     0.95  |     0.82  |
> | Cosine-MiniLM vs LLaMA   |     0.94  |     0.83  |
> | W2V vs LLaMA       |     0.93  |     0.87  |
>
> **Table 2: Performance of CPQ with Different Clustering Methods on TriviaQA (LLaMA-3-8B Generations)**
> (*All std < 0.05*)
> | Clustering Method   | 1 - α | Coverage | EE Fraction |
> |:--------------------|:-----:|:---------:|:------------:|
> | RoBERTa             | 0.8   | 0.81      | 0.42         |
> |                     | 0.6   | 0.61      | 0.09         |
> | W2V                 | 0.8   | 0.81      | 0.44         |
> |                     | 0.6   | 0.62      | 0.10         |
> | Cosine-MiniLM       | 0.8   | 0.82      | 0.44         |
> |                     | 0.6   | 0.62      | 0.09         |
> | LLaMA (default)     | 0.8   | 0.80      | **0.42**     |
> |                     | 0.6   | 0.61      | **0.08**     |
>
> **our key findings that address your questions:**
> - In our ablation study, switching between **different LLMs and clustering methods yielded comparable performance and CP metrics; no dramatic shifts were observed.**
> - Unsupervised **methods like Word2Vec + KMeans were feasible in our experiments.** Similarly the **impact on downstream CP metrics was minor and comparable to that of LLM-based clustering.**
>
> These results suggest that as long as semantically similar outputs are grouped reasonably well, the specific clustering method has little effect on overall performance. That said, clustering remains an active research area, especially for very long‐form or domain-specific generations, and users can leverage advances in NLP to fit their particular use cases
>
>
> ## **Imperfect Oracles and Costs**
> We wanted to clarify that although the two principles are derived in a theoretical framework that assumes perfect oracles, our finite sample algorithm is designed for any (imperfect) oracle/LLM. Subsequently, all of our experiments are run with real-world, black-box LLMs. These models are inherently imperfect. The results show that the population-level principles lead to significant improvements in practice.
>
> As for computational costs, our method is comparable to techniques like best-of-(N) sampling, widely used in practice, which query the model multiple times and select the best response. Our framework aligns with this paradigm but offers a principled way to decide how many queries to issue and when to stop. In high-stakes settings like medical decision-making, where decisions often take at least several minutes, this modest overhead is justified by the added reliability. Inference time for 10–20 queries typically ranges from a few seconds (small models) to a couple minutes (large models).
>
> **Practicality in very low query regimes**:  We provided a more detailed discussion on this setting in our response to Reviewer tHYJ.
> **When the query budget is very limited (e.g., ≤ 5), Principle 2 can still be used on its own and yield meaningful gains.**. Unlike Principle 1, which requires $N_2$ (doubletons), Principle 2 only depends on $N_1$ (singletons) via the Good–Turing estimator, which performs well even with few samples.
>
> To illustrate this, we ran **an experiment with a fixed query budget of 5 per test point** (non-adaptive), applying only the calibration step from Principle 2.
>
> ## **Fairness of Baseline Comparisons**
> As the reviewer also noted, our work addresses a previously unsolved challenge in conformal prediction for generative models, namely, how to manage the three-way trade-off between coverage, test-time query budget, and the informativeness of prediction sets. Given the novelty of this problem, it is natural that no existing baseline fully aligns with the setting we aim to solve. That said, we took two steps to strengthen the robustness of our experimental evaluation.
>
> 1. We conducted fine-grained ablation studies to isolate and assess the contribution of each algorithmic principle to the informativeness of the resulting prediction sets.
> 2. We adapted, using the best available techniques, two recent conformal prediction methods that share our high-level objective: constructing valid prediction sets from LLM outputs. We also carefully tuned the query budgets to be favorable to these baseline methods, ensuring that the comparison remains as fair as possible.
>
> ## **Other Questions:**
> Q2: Our framework, based on two derived principles, leverages the fact that the decoupled optimization admits closed-form solutions. In contrast, the joint optimization is significantly more complex: it lacks structural properties such as convexity and, to our knowledge, does not admit a closed-form solution. This makes it unclear how to design a practical finite-sample algorithm for the joint objective. Although the exact performance gap is difficult to quantify due to this intractability, our empirical results show that the decoupled approach performs strongly across tasks and models.
>
> Q3: Distribution shift is an active research area in the CP community, extending beyond LLM applications. In this work, we focused on CP challenges specific to generative modeling. That said, robustness to domain shifts is an important direction, which we will include in the future work section of the revised manuscript. As a starting point, one could integrate ideas from [1], e.g. calibrating a linear head on pretrained prompt representations via quantile regression instead of a scalar threshold calibration, potentially improving robustness. However, combining this with our missing mass framework remains an open question.
>
> [1]: Conformal prediction with conditional guarantees, Gibbs et. al.
>
> Q5: While we do not address p-value computation, there are related efforts in the literature that leverage conformal ideas to construct empirical p-values, e.g. in the context of outlier detection and multiple testing (e.g., Bates et al., 2021; Jin et al., 2022).
> Extending such ideas to our setting would require a careful treatment of the EE label and the missing mass, which introduce additional layers of uncertainty not present in standard CP, which is a promising direction for future research.

---

### Official Review · Reviewer_ZMjq · 2025-07-03

**Clarity:** 3
**Significance:** 3
**Originality:** 3
**Rating:** 5
**Confidence:** 3

**Summary:**

This paper studies conformal prediction for LLMs. It focuses on a setting with only query access to black-box LLMs. The authors propose Conformal Prediction with Query Oracle (CPQ), a framework that balances coverage, informativeness, and query budget. Specifically, CPQ consists of two stages. First, a query budget is determined based on a missing-mass estimator. The LLM is then queried for that specific number of times, and a final prediction set is constructed from the sampled responses by thresholding a conformity score. Experiments on three real-world benchmarks demonstrate the effectiveness of the proposed method.

**Questions:**

Please see weaknesses.

**Ethical Concerns:**

["NO or VERY MINOR ethics concerns only"]

**Final Justification:**

The authors' response has addressed my issues, so I have raised my score.

**Limitations:**

Yes

**Quality:**

3

**Strengths And Weaknesses:**

Strengths
1. The paper is well written. Although involving many technical contents, it achieves a good balance between including enough details and background for understanding, as well as deferring detailed proof to the appendix. The practical algorithm is also well motivated by the theoretical analyses.
2. Experiments show that the method has a smaller EE fraction, demonstrating the informativeness of the prediction set.

Weaknesses
1. All three datasets focus more on the reasoning tasks. It would be better if the authors could demonstrate the effectiveness on other tasks, e.g., factual QA.
2. Results in Figure 1 are a bit difficult to interpret, especially on the size of the prediction set, which first increases and then decreases. My understanding is that it's because of EE fraction. But it's hard to compare across different methods when the two metrics entangle.

---

> ### Author Rebuttal · Authors · 2025-07-28
>
> We thank the reviewer for their thoughtful feedback. We're glad you found the paper well written, clearly organized, and supported by sound experimental evidence. Below we respond to each of the concerns you raised.
>
> ### **Evaluation beyond reasoning benchmarks (e.g., factual QA):**
>
> Thank you for this suggestion. We also agree adding other types of datasets (non-reasoning) can enrich our experimental demonstration. In response, we conducted additional experiments on factual QA tasks. These new experiments include:
> - **TriviaQA**, a widely used open-domain factual QA benchmark.
> - **SimpleQA**, an adversarial dataset where even GPT-4o achieves only <~30% EM accuracy in our experiments.
>
> Additionally, for these new datasets, we conduct experiments on larger black-box models, including **GPT-3.5-turbo** and **GPT-4o**, in addition to LLaMA-3-8B, which was used in the experiments already reported in the paper.
>
> **Across all settings, we observed consistent trends with the results presented in the main paper**. Specifically, our method consistently improves the EE fraction when such improvement is feasible (i.e., at moderate coverage levels), and yields smaller average set sizes at higher coverage levels where reducing the EE fraction becomes increasingly difficult. For a more detailed discussion on how to interpret the trade-off between EE fraction and set size, please refer to the section **Interpreting Figure 1** toward the end of this rebuttal.
>
> In what follows, we provide a summary of these new results, and we will add a more detailed report in the revised version.
> #### **SIMPLE QA – GPT-4o**
>
> | Algorithm           | 1 - α | Emp Cov ± Std    | EE ± Std        | Avg Set Size ± Std |
> |:--------------------|:-----:|------------------:|-----------------:|--------------------:|
> | Vanilla             | 0.60  | 0.62 ± 0.04       | 0.16 ± 0.03      | 0.85 ± 0.02         |
> | P1                  | 0.60  | 0.59 ± 0.06       | 0.11 ± 0.03      | 1.20 ± 0.14         |
> | P1 + P2 (CPQ)       | 0.60  | 0.61 ± 0.05       | **0.04 ± 0.04**  | 3.47 ± 0.53         |
> | ---                 |       |                   |                  |                     |
> | Vanilla             | 0.65  | 0.65 ± 0.06       | 0.19 ± 0.05      | 0.89 ± 0.07         |
> | P1                  | 0.65  | 0.63 ± 0.06       | 0.16 ± 0.05      | 1.28 ± 0.14         |
> | P1 + P2 (CPQ)       | 0.65  | 0.65 ± 0.05       | **0.11 ± 0.06**  | 2.89 ± 0.67         |
> | ---                 |       |                   |                  |                     |
> | Vanilla             | 0.70  | 0.68 ± 0.07       | 0.23 ± 0.06      | 0.89 ± 0.03         |
> | P1                  | 0.70  | 0.68 ± 0.05       | 0.22 ± 0.04      | 1.04 ± 0.04         |
> | P1 + P2 (CPQ)       | 0.70  | 0.69 ± 0.04       | **0.18 ± 0.06**  | 1.93 ± 0.58         |
> | ---                 |       |                   |                  |                     |
> | Vanilla             | 0.80  | 0.82 ± 0.06       | 0.48 ± 0.10      | 1.15 ± 0.21         |
> | P1                  | 0.80  | 0.79 ± 0.06       | 0.41 ± 0.11      | 1.77 ± 0.80         |
> | P1 + P2 (CPQ)       | 0.80  | 0.79 ± 0.05       | **0.40 ± 0.09**  | 0.70 ± 0.18         |
>
> #### **TRIVIAQA – GPT-3.5-Turbo**
>
> | Algorithm           | 1 - α    | Emp Cov ± Std    | EE ± Std        | Avg Set Size ± Std |
> |:--------------------|:----:|------------------:|-----------------:|--------------------:|
> | Vanilla             | 0.60 | 0.58 ± 0.06       | 0.08 ± 0.04      | 1.61 ± 0.31         |
> | P1                  | 0.60 | 0.58 ± 0.06       | 0.08 ± 0.04      | 1.52 ± 0.38         |
> | P1 + P2 (CPQ)       | 0.60 | 0.60 ± 0.07       | **0.01 ± 0.01**  | 5.34 ± 1.36         |
> | ---                 |      |                   |                  |                     |
> | Vanilla             | 0.65 | 0.63 ± 0.06       | 0.15 ± 0.05      | 1.76 ± 0.32         |
> | P1                  | 0.65 | 0.64 ± 0.06       | 0.14 ± 0.05      | 1.68 ± 0.30         |
> | P1 + P2 (CPQ)       | 0.65 | 0.64 ± 0.06       | **0.06 ± 0.05**  | 5.42 ± 1.09         |
> | ---                 |      |                   |                  |                     |
> | Vanilla             | 0.70 | 0.69 ± 0.06       | 0.23 ± 0.06      | 1.54 ± 0.28         |
> | P1                  | 0.70 | 0.68 ± 0.06       | 0.22 ± 0.07      | 1.43 ± 0.17         |
> | P1 + P2 (CPQ)       | 0.70 | 0.69 ± 0.07       | **0.15 ± 0.09**  | 3.90 ± 1.54         |
> | ---                 |      |                   |                  |                     |
> | Vanilla             | 0.80 | 0.78 ± 0.05       | 0.38 ± 0.07      | 2.35 ± 0.57         |
> | P1                  | 0.80 | 0.78 ± 0.06       | 0.40 ± 0.09      | 2.46 ± 0.65         |
> | P1 + P2 (CPQ)       | 0.80 | 0.79 ± 0.07       | **0.39 ± 0.07**  | 1.05 ± 0.52         |
>
>
> #### **TRIVIAQA – LLaMA3-8B**
>
> | Algorithm           | 1 - α    | Emp Cov ± Std    | EE ± Std       | Avg Set Size ± Std |
> |:--------------------|:----:|------------------:|----------------:|--------------------:|
> | Vanilla             | 0.60 | 0.61 ± 0.02       | 0.21 ± 0.01     | 0.82 ± 0.05         |
> | P1                  | 0.60 | 0.61 ± 0.03       | 0.16 ± 0.03     | 1.30 ± 0.07         |
> | P1 + P2 (CPQ)       | 0.60 | 0.60 ± 0.03       | **0.10 ± 0.03** | 2.99 ± 0.36         |
> | ---                 |      |                   |                 |                     |
> | Vanilla             | 0.65 | 0.67 ± 0.05       | 0.30 ± 0.05     | 0.83 ± 0.12         |
> | P1                  | 0.65 | 0.66 ± 0.02       | 0.25 ± 0.02     | 1.17 ± 0.08         |
> | P1 + P2 (CPQ)       | 0.65 | 0.66 ± 0.04       | **0.21 ± 0.05** | 1.93 ± 0.32         |
> | ---                 |      |                   |                 |                     |
> | Vanilla             | 0.70 | 0.70 ± 0.03       | 0.34 ± 0.04     | 0.81 ± 0.05         |
> | P1                  | 0.70 | 0.71 ± 0.04       | 0.30 ± 0.03     | 1.17 ± 0.05         |
> | P1 + P2 (CPQ)       | 0.70 | 0.71 ± 0.04       | **0.28 ± 0.03** | 1.39 ± 0.16         |
> | ---                 |      |                   |                 |                     |
> | Vanilla             | 0.80 | 0.81 ± 0.04       | 0.52 ± 0.06     | 1.02 ± 0.03         |
> | P1                  | 0.80 | 0.80 ± 0.04       | 0.46 ± 0.05     | 1.37 ± 0.14         |
> | P1 + P2 (CPQ)       | 0.80 | 0.80 ± 0.04       | 0.46 ± 0.05     | 0.62 ± 0.10         |
>
> ### **Interpreting Figure 1**
> we’re happy to clarify the trends in Figure 1 and how to interpret the interaction between the EE fraction and the average prediction set size.
> The average set size reported in the figure refers to the average number of explicit labels (excluding EE) included in the prediction set. When EE is included, it signifies complete uncertainty over the remainder of the label space, effectively making the set size infinite.
>
> Thus, when comparing two methods, one should first examine the **EE fraction** (the smaller, the better). If the EE fractions are similar (which often occurs at higher coverage levels), then the **average set size** becomes the distinguishing factor (again, smaller is better). This trade-off between EE and set size is precisely what our algorithmic framework is designed to capture.
>
> At moderate coverage levels, where EE labels can potentially be avoided, our method significantly reduces the fraction of EE in the prediction sets. At higher coverage levels, where including EE becomes unavoidable to guarantee coverage, our method instead focuses on reducing the average set size, making the non-EE prediction sets more practical and informative.
>
> This behavior is enabled by the careful design of the two key principles discussed in the paper. It is also reflected in the plots: as we sweep the value of $(1 - \alpha)$ from low to high, the set size initially increases (to avoid using EE), then decreases (as including EE becomes necessary and the method adapts).
>
> In other words, in the low-$(1 - \alpha)$ regime, our method outperforms baselines by reducing EE inclusion. In the high-$(1 - \alpha)$ regime, where all methods must include EE frequently, the advantage shifts to smaller prediction sets, while keeping the number of EE labels fixed.
>
> We thank the author again for bringing this matter up, and we will revise the caption and text around Figure 1 to make sure this is clearly communicated in the revised version.

---

### Comment · Area_Chair_URyg · 2025-08-05

Dear Reviewers, given the authors' response, if you have not done so, please raise any remaining questions and/or concerns in a timely fashion so the authors have a chance to reply.

I remind you that Reviewers must participate in discussions with authors before submitting "Mandatory Acknowledgement”.

Thank you for your work.

---

### Note · Authors · 2025-08-14

We thank all reviewers for their thoughtful feedback and kind recognition of the paper’s strengths; including its clear writing, novel framework, useful theoretical lens, and strong experimental grounding. During the rebuttal, we:

- **Expanded evaluations** to larger models (GPT-4o, GPT-3.5-turbo) and new datasets (TriviaQA, SimpleQA) to test beyond reasoning tasks and under extreme hallucination scenarios.
- **Analyzed multiple clustering methods**, showing our approach remains robust across various clustering mechanisms.
- **Demonstrated effectiveness in low-query regimes**, by showing the effectiveness of principle 2 in isolation using fixed 5 queries.

We also addressed additional reviewer questions and are glad to see that Reviewer YqEn found our responses satisfactory and raised their score.

---

### Decision · Program_Chairs · 2025-09-17

**Decision:**

Accept (poster)

**Comment:**

This paper proposes a new contribution on conformal prediction for large language models (LLM).  In such a setting, only a few outputs are sampled from the generative model (a few answers to a prompt for a LLM for instance). Therefore, estimating the uncertainty about the missing mass i.e. about the fact that the correct output is unseen is very challenging.  The authors propose a complete framework to formally capture the trade-off between coverage, informativeness, and query budget consisting of a query policy and a set map and split the problem into two steps:  allocating queries across covariate points and  constructing prediction sets with desired coverage. The proposed approach relies on simple yet theoretically grounded principles which allows to design a finite-sample algorithm outperforming recent conformal prediction baselines for generative models.

All reviewers agreed on the interest of the paper for the ML community, and noted that the paper is clearly written on an important problem about the limitations of conformal predictions for generative models. Although, they had questions on the experiments, they were mainly convinced by the experimental section of the paper.

The reviewers had mainly concerns on the experiments, in particular on the variety of the tasks considered in the experiments and on the baselines used by the authors. During the rebuttal the authors conducted additional experiments with TriviaQ and SimpleQA, using additional large LLM (GPT-3.5-turbo and GPT-4o). The results of these experiments confirm the numerical claims of the original paper. They also conducted an ablation using fixed querying strategy to highlight the performance of their optimal calibration rule.

A revised version of the paper including the additional experiments and detailed comments provided to the reviewers (detailed baselines, revised figures, discussions on the practical costs) will make a contribution of interest for a large ML audience.